# Beyond Graphs: Can Large Language Models Comprehend Hypergraphs?

**Yifan Feng[1], Chengwu Yang[2], Xingliang Hou[3], Shaoyi Du[2], Shihui Ying[4], Zongze Wu[5*], Yue Gao[1] ***

[1]School of Software, BNRist, THUIBCS, BLBCI, Tsinghua University
[2]Institute of Artificial Intelligence and Robotics, College of Artificial Intelligence, Xi'an Jiaotong University
[3]School of Software, Xi'an Jiaotong University
[4]Department of Mathematics, School of Science, Shanghai University
[5]College of Mechatronics and Control Engineering, Shenzhen University
evanfeng97@gmail.com, slamdunkycw@gmail.com
HouXL@stu.xjtu.edu.cn, dushaoyi@xjtu.edu.cn
shying@shu.edu.cn, zzwu@gdut.edu.cn, gaoyue@tsinghua.edu.cn

## Abstract

Existing benchmarks like `NLGraph` and `GraphQA` evaluate LLMs on graphs by focusing mainly on pairwise relationships, overlooking the high-order correlations found in real-world data. Hypergraphs, which can model complex beyond-pairwise relationships, offer a more robust framework but are still underexplored in the context of LLMs. To address this gap, we introduce `LLM4Hypergraph`, the first comprehensive benchmark comprising 21,500 problems across eight low-order, five high-order, and two isomorphism tasks, utilizing both synthetic and real-world hypergraphs from citation networks and protein structures. We evaluate six prominent LLMs, including GPT-4o, demonstrating our benchmark's effectiveness in identifying model strengths and weaknesses. Our specialized prompting framework incorporates seven hypergraph languages and introduces two novel techniques, *Hyper-BAG* and *Hyper-COT*, which enhance high-order reasoning and achieve an average 4% (up to 9%) performance improvement on structure classification tasks. This work establishes a foundational testbed for integrating hypergraph computational capabilities into LLMs, advancing their comprehension. The source codes are at https://github.com/iMoonLab/LLM4Hypergraph.

## 1 Introduction

Large Language Models (LLMs) (Vaswani, 2017; Devlin, 2018; Brown, 2020; Ouyang et al., 2022) have made significant strides in domains such as dialogue systems (Bubeck et al., 2023) and image understanding (Zhao et al., 2023). However, they often produce untruthful or unsupported content, known as *hallucinations* (Wang et al., 2023). To mitigate this, Retrieval-Augmented Generation (RAG) (Vu et al., 2023) enhances prompts with relevant, factual, and up-to-date information (Khandelwal et al., 2019), thereby grounding outputs more effectively. RAG typically retrieves structured data with complex relational dependencies (Guu et al., 2020), such as social networks or molecular structures, which are efficiently represented as graphs. Graph representations capture intricate interdependencies and provide a concise encapsulation of data relationships. This has spurred research to improve LLMs' understanding of graph-structured data (Guo et al., 2023), leading to benchmarks like `NLGraph` (Wang et al., 2024), `GraphQA` (Fatemi et al., 2023), and `LLM4DyG` (Zhang et al., 2023). These benchmarks evaluate and enhance LLMs' capabilities in handling graph-related tasks, promoting the integration of graph-based representations in large language models.

However, real-world data often involve complex correlations beyond simple pairwise relationships (Zhou et al., 2006). For example, sentences within a document sharing common keywords may exhibit high-order correlations that traditional graph models fail to capture (PM et al., 2017). In multimodal scenarios (Kim et al., 2020; Feng et al., 2023), interactions across different data types further increase correlation complexity, exceeding the capabilities of conventional graphs, which

---
*Corresponding authors: Yue Gao and Zongze Wu

are limited to pairwise correlations. In contrast, hypergraphs can model high-order correlations, capturing intricate interdependencies among multiple entities simultaneously. This capability allows hypergraphs to more accurately represent the complex relationships in retrieved information, thereby enhancing prompting strategies in LLMs. Despite these advantages, hypergraphs pose a challenge for LLMs, which are primarily designed to process linear textual data and cannot naturally ingest hypergraph structures (Feng et al., 2024). This limitation hinders the integration of hypergraph-based representations into LLM training and inference. *Therefore, investigating whether LLMs can comprehend hypergraphs and developing prompts to improve their understanding of hypergraph structures remains an underexplored and promising research direction.*

Here, we introduce the `LLM4Hypergraph` Benchmark, the first comprehensive testbed designed to evaluate LLMs' understanding and reasoning of hypergraphs. Unlike existing graph benchmarks that focus on low-order, pairwise vertex correlation tasks, our benchmark presents high-order tasks involving vertex sets and isomorphism tasks that assess models' ability to recognize and interpret global structural correlations within hypergraphs. Our benchmark includes both synthetic and real-world hypergraphs, such as citation and protein networks, ensuring diverse and representative evaluations across domains. It comprises 21,500 problems, covering eight low-order tasks, five high-order tasks, and two isomorphism tasks. Low-order tasks extend pairwise correlation reasoning to more complex vertex interactions, while high-order tasks involve reasoning about relationships among vertex sets, capturing the rich multi-vertex correlations inherent to hypergraphs. Isomorphism tasks evaluate the models' ability to identify and interpret global structural patterns and symmetries within hypergraphs. To assess our benchmark's effectiveness, we evaluate six mainstream LLMs (including ERNIE-Lite-8K, ERNIE-Speed-128K, Qwen-Long, LLaMA3-8B, GPT-3.5-Turbo, and GPT-4o) on the `LLM4Hypergraph` Benchmark, as shown in Figure 1. The evaluation revealed four findings:

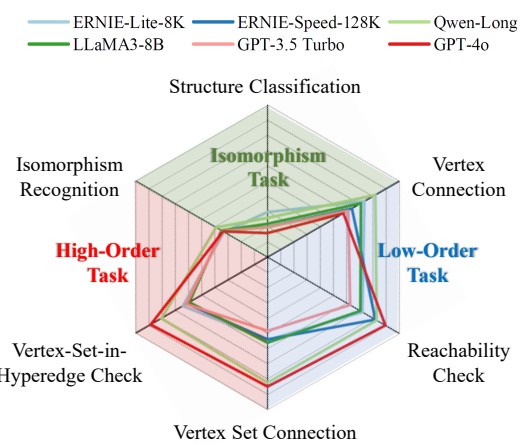

Figure 1: Performance of six leading LLMs on three types of hypergraph understanding task.

- In-context learning enhances LLMs' comprehension and performance in hypergraph tasks.
- Current mainstream LLMs struggle with isomorphism recognition, especially in larger hypergraphs, though improvements are possible.
- The proposed High-order languages (specified for hypergraphs) outperform low-order languages in enabling LLMs to understand beyond-pairwise correlations in hypergraphs.
- The proposed Hyper-COT and Hyper-BAG significantly boost LLMs' performance on more challenging hypergraph tasks like structure classification.

Additionally, we design a specialized prompting framework for hypergraphs with seven languages for low-order and high-order structures, enabling nuanced interactions with LLMs. We introduce two instruction-based prompting techniques: *Hyper-BAG* and *Hyper-COT*, which guide LLMs to visualize high-order correlations and better understand multi-vertex relationships. Experimental results show that Hyper-COT improves performance by 4% on average (up to 9%) in challenging structure classification tasks. Despite these advancements, challenges in high-order reasoning and encoding remain. Our work paves the way for future research to overcome these limitations and further integrate hypergraph capabilities into LLMs.

## 2 LLM4HYPERGRAPH BENCHMARK

In this section, we introduce the `LLM4Hypergraph` Benchmark, designed to evaluate LLMs' ability to comprehend the intricate higher-order structures of hypergraphs. To ensure a comprehensive assessment, the benchmark encompasses a diverse array of tasks and hypergraph configurations, as shown in Figure 2. It includes **Low-order tasks** (basic hypergraph properties and pairwise rela-

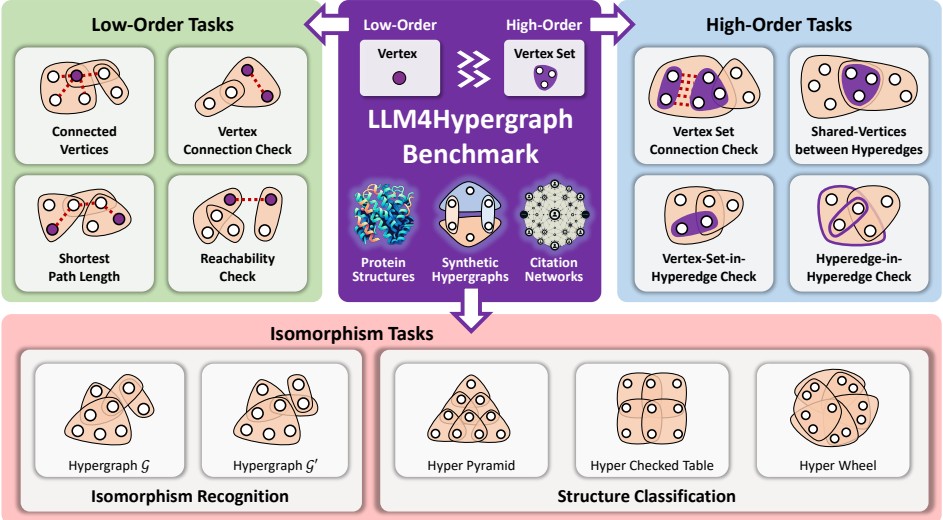

Figure 2: Overview of the LLM4Hypergraph Benchmark.

tional inferences), **High-order tasks** (understanding complex multi-vertex interactions and higher-order dependencies), and **Isomorphic tasks** (recognizing structural equivalences despite hypergraph variations). Additionally, the `LLM4Hypergraph` Benchmark incorporates hypergraphs of varying scales and types, ranging from small-scale hypergraphs with limited entities and relationships to large-scale, intricate hypergraphs that emulate real-world complexity. By addressing these dimensions, the benchmark provides a holistic framework for assessing the versatility and depth of LLMs in comprehending and processing hypergraphs. Our `LLM4Hypergraph` Benchmark comprises a diverse collection of hypergraphs, including both synthetic and real-world instances. Synthetic hypergraphs are generated using random and regular-structured methods, while real-world hypergraphs are sourced from citation and protein networks. These hypergraphs are categorized by scale to ensure a balanced complexity for comprehensive evaluation. For a detailed description of the hypergraph generation process, please refer to Appendix D.

## 2.1 TASK DESIGN

In this subsection, we briefly introduce the 15 tasks included in our `LLM4Hypergraph` Benchmark. Detailed descriptions and methodologies for these tasks are provided in Appendix E.

**Isomorphism Tasks.** Isomorphic tasks are a fundamental category within the `LLM4Hypergraph` Benchmark, designed to evaluate LLMs' ability to recognize structural equivalences and classify hypergraphs based on their overall architectural patterns. These tasks are crucial for applications in specialized fields such as molecular and protein structure analysis, where distinguishing between subtly different structural configurations is essential. We introduce two key isomorphic tasks:

- *Isomorphism Recognition.* This task assesses the model's ability to determine whether two hypergraph representations correspond to the same underlying structure.
- *Structure Classification.* This task evaluates the model's proficiency in distinguishing hypergraphs based on their macro-level architectural frameworks.

**Low-Order Tasks.** In addition to isomorphic tasks, our `LLM4Hypergraph` Benchmark incorporates a series of low-order tasks designed to test LLMs' understanding of fundamental hypergraph properties and vertex relationships. These tasks focus on basic hypergraph attributes and simple connectivity patterns essential for more complex structural analyses.

- *Hyperedge Count.* Counts the total number of hyperedges.
- *Vertex Count.* Counts the total number of vertices.
- *Vertex Degree.* Counts the hyperedges connected to a specific vertex.
- *Vertex Connection.* Checks if two vertices are directly connected by a hyperedge.
- *Connected Vertices.* Lists all vertices connected to a given vertex.

Table 1: Statistics of the LLM4Hypergraph Benchmark.

| Task | Task Name | Task Type | Difficulty | #Sample |
|---|---|---|---|---|
| Low-Order | Hyperedge Count | Counting Problems | Easy | 1500 |
| | Vertex Count | Counting Problems | Easy | 1500 |
| | Vertex Degree | Counting Problems | Easy | 1500 |
| | Vertex Connection Check | Decision Problems | Medium | 1500 |
| | Reachability Check | Decision Problems | Medium | 1500 |
| | Shortest Path | Computational Problems | Hard | 1500 |
| | Connected Vertices | Descriptive Problems | Hard | 1500 |
| | Disconnected Vertices | Descriptive Problems | Hard | 1500 |
| High-Order | Hyperedge Degree | Counting Problems | Easy | 1500 |
| | Vertex Set Connection Check | Decision Problems | Medium | 1500 |
| | Vertex-Set-in-Hyperedge Check | Decision Problems | Medium | 1500 |
| | Hyperedge-in-Hyperedge Check | Decision Problems | Medium | 1500 |
| | Shared-Vertices between Hyperedges | Descriptive Problems | Hard | 1500 |
| Isomorphism | Isomorphism Recognition | Computational Problems | Hard | 1500 |
| | Structure Classification | Classification Problems | Hard | 500 |
| Total | 15 | 5 | 3 | 21,500 |

- *Disconnected Vertices.* Lists all vertices not connected to a given vertex.
- *Shortest Path.* Finds the shortest path between two vertices.
- *Reachability Check.* Determines if one vertex can be reached from another.

**High-Order Tasks.** Building upon low-order tasks, `LLM4Hypergraph` Benchmark introduces high-order tasks designed to assess LLMs' comprehension of complex correlations between vertex sets within hypergraphs. Unlike low-order tasks that focus on individual vertices and hyperedges, high-order tasks evaluate the model's ability to understand and manipulate relationships involving groups of vertices and their interactions through hyperedges.

- *Hyperedge Degree.* Determines the number of vertices in a given hyperedge.
- *Vertex Set Connection (VS Connection).* Checks if two vertex sets are jointly contained within at least one hyperedge.
- *Vertex-Set-in-Hyperedge Check (VS-in-He Check).* Determines if a set of vertices is entirely contained within any hyperedge.
- *Hyperedge-in-Hyperedge Check (He-in-He Check).* Assesses if one hyperedge is completely contained within another hyperedge.
- *Shared-Vertices between Hyperedges.* Identifies and outputs the set of vertices shared between two hyperedges.

## 2.2 BENCHMARK STATISTICS

Our `LLM4Hypergraph` Benchmark comprises 14 tasks totaling 21,500 problems, as detailed in Table 1. These tasks are categorized into five types: Counting Problems (counting specific elements), Computational Problems (numerical answers), Decision Problems (yes/no responses), Descriptive Problems (listing sets of vertices or hyperedges), and Classification Problems (categorical labels). Each type includes 1,500 samples, providing a comprehensive assessment of LLMs' abilities in numerical computations, binary decisions, descriptive generation, and hypergraph classification.

## 3 PROMPT ENGINEERING FOR HYPERGRAPHS

Prompt design is essential for accurately evaluating LLMs in our `LLM4Hypergraph` Benchmark, as prompts guide the models' understanding of hypergraph structures. Here, we integrate existing strategies such as CoT (Wei et al., 2022; Kojima et al., 2022) and Few-Shot Prompting (Brown, 2020; Zhou et al., 2022a) to develop a prompt framework tailored for hypergraph-related tasks. Additionally, we introduce a hypergraph language for textual descriptions and adapt these techniques with modifications like Hyper-BAG and Hyper-COT to better accommodate hypergraphs.

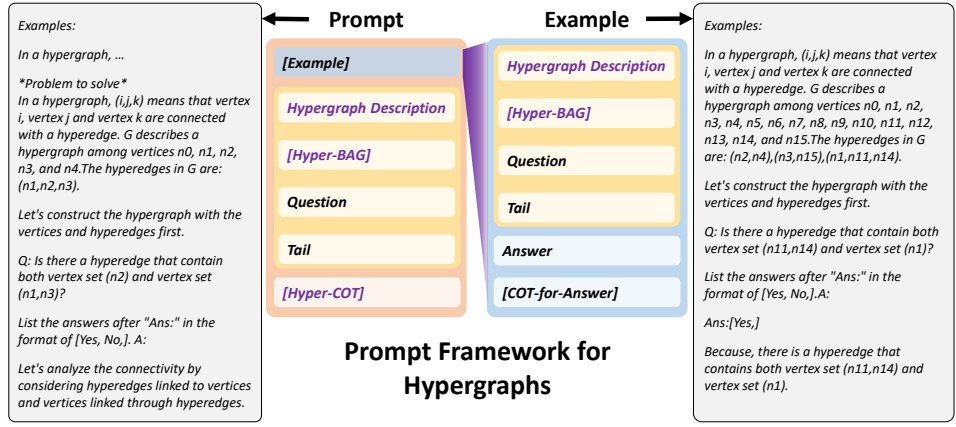

Figure 3: The proposed prompt structure for hypergraphs.

## 3.1 PROMPT FRAMEWORK

Prompt design is crucial for accurately evaluating LLMs in our `LLM4Hypergraph` Benchmark, as it directs the models' understanding of hypergraph structures. Figure 3 illustrates our prompt framework, which consists of six components: *Example*, *Hypergraph Description*, *Hyper-BAG*, *Question*, *Tail*, and *Hyper-COT*. The *Example* section includes question-answer pairs and may optionally contain *COT-for-Answer* for detailed reasoning. Optional modules are indicated by brackets in the figure. More details of the framework are provided in Appendix F. In the following, we briefly introduce the following prompt configurations:

- *ZERO-SHOT*: Includes only *Hypergraph Description*, *Question*, and *Tail*, without examples or additional reasoning prompts.
- *ZERO-HYPER-COT*: Extends Zero-Shot by adding *Hyper-COT* after the *Tail*, introducing hypergraph-specific reasoning.
- *FEW-SHOT*: Adds *Example* content with *Hypergraph Description*, *Question*, *Tail*, and *Answer* to provide concrete instances for guidance.
- *COT*: Enhances Few-Shot by including *COT-for-Answer* in each example, offering detailed reasoning steps for more accurate responses.
- *COT-HYPER-BAG*: Builds on the CoT setup by inserting *Hyper-BAG* between *Hypergraph Description* and *Question*, contextualizing the model's focus on hypergraph construction and comprehension.

## 3.2 HYPERGRAPH LANGUAGE

Unlike traditional graphs, hypergraphs allow hyperedges to connect any number of vertices, requiring more complex textual descriptions. Abstract high-order descriptions may hinder LLM comprehension, while low-order descriptions enhance understanding but omit essential information. To address this, we designed two hypergraph languages: *Low-Order Structure Language* and *High-Order Structure Language*, as shown in Figure 4. The former focuses on pairwise relationships, while the latter preserves complex multi-vertex correlations. More details can refer to Appendix G.

**Low-Order Structure Language.** To facilitate LLMs' understanding of hypergraph structures through pairwise correlations, we introduce three low-order structure languages as follows:

- *Low-Order-Incidence Description (LO-Inc)*: Describes pairwise connections between vertices, e.g., "Vertex $v_1$ is connected to vertices $v_2$ and $v_3$."
- *Neighbor-Pair Description (N-Pair)*: Lists all pairs of vertices that share a hyperedge, e.g., "$(v_1, v_2), (v_1, v_3)$."
- *Raw Adjacency Matrix Description (Adj-Mat)*: Uses a numerical adjacency matrix where binary values indicate connections between vertex pairs.

**High-Order Structure Language**. To further enhance LLMs' comprehension of hypergraph structures through higher-order correlations, we introduce four high-order structure languages as follows:

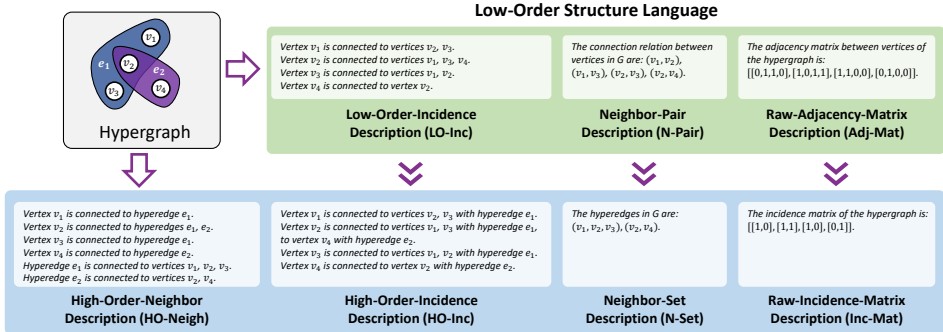

Figure 4: The proposed "Hypergraph Language" for *Hypergraph Description* section.

- *High-Order Neighbor Description (HO-Neigh)*: Describes hypergraph relationships in two stages, detailing connections between vertices and hyperedges, then hyperedges and their vertices.

- *High-Order Incidence Description (HO-Inc)*: Extends LO-Inc by including higher-order correlations, such as "Vertex $v_1$ is connected to vertices $v_2$ and $v_3$ with hyperedge $e_1$."

- *Neighbor-Set Description (N-Set)*: Lists entire sets of vertices connected by each hyperedge, for example, "$(v_1, v_2, v_3)$."

- *Raw Incidence Matrix Description (Inc-Mat)*: Uses an incidence matrix where each entry indicates the inclusion of a vertex in a hyperedge.

## 3.3 SPECIFICAL PROMPTING FOR HYPERGRAPHS

Given that hypergraphs are difficult to describe using conventional language, we introduce two specialized prompting techniques to enhance LLMs' comprehension of hypergraph structures: *Hyper-BAG* and *Hyper-COT*. More details can be found in Appendix H.

- *Build-a-Hypergraph Prompting (Hyper-BAG)* facilitates mental visualization of hypergraphs by guiding LLMs to imagine their architecture, helping the model form a coherent representation of hypergraph connections (Wang et al., 2024).

- *Chain-of-Thought for Hypergraphs (Hyper-COT)* adapts traditional CoT prompting by incorporating step-by-step reasoning tailored for hypergraphs. It includes specific prompts that guide the model to break down complex hypergraph tasks into manageable steps:
  - v1: *"Let's think step by step. Make sure the data is calculated and recorded accurately at each step."*
  - v2: *"Let's analyze the connectivity by considering hyperedges linked to vertices and vertices linked through hyperedges."*
  - v3: *"Let's think hyperedges connected by vertices then vertices connected by hyperedges."*

## 4 EXPERIMENTS

Based on the `LLM4Hypergraph` Benchmark, we aim to investigate *whether language models can comprehend hypergraphs* by evaluating LLMs and different prompting settings.

**Experimental Settings.** We evaluate six LLMs: ERNIE-Lite-8K (Zhang et al., 2019), ERNIE-Speed-128K (Sun et al., 2021), Qwen-Long (Bai et al., 2023), LLaMA3-8B (Touvron et al., 2023), GPT-3.5-Turbo (Brown, 2020), and GPT-4 (Achiam et al., 2023), selected for their diverse architectures. In our experiments, we employ Qwen-Long as the LLM due to its robust performance and scalability. In the "Few-Shot", "CoT", and "CoT-Hyper-BAG" settings, we provide two examples by default. For Decision Problems, we maintain a balanced positive-to-negative ratio of 1:1 to ensure fairness. Other tasks assess consistency by comparing LLM outputs to ground truth.

## 4.1 RESULTS OF DIFFERENT HYPERGRAPH LANGUAGES

We evaluated different hypergraph languages under various settings using synthetic hypergraphs, with the experimental results presented in Table 2 and Table 3.

**Observation 1: Low-order structure languages enhance LLMs' understanding of vertex correlations within hypergraphs.** As illustrated in Table 2, in low-order tasks, high-order languages outperform low-order languages in simple tasks, achieving higher accuracy due to their ability to accurately describe high-order structures. However, in medium and difficult low-order tasks, low-order languages demonstrate superior performance. This decline in high-order language effectiveness in more challenging low-order tasks can be attributed to the redundancy of high-order descriptions when the primary focus shifts to investigating pairwise vertex correlations. Consequently, when the objective is to examine relationships between individual vertices within hypergraphs, low-order languages are more beneficial for enhancing LLMs' understanding.

Table 2: Results of different prompting frameworks and hypergraph languages.

| Difficulty | | Esay | | | Medium | | Hard | | |
|---|---|---|---|---|---|---|---|---|---|
| Settings | | Hyperdge Count | Vertex Count | Vertex Degree | Vertex Connection | Reachability Check | Shortest Path | Connected Vertices | Disconnection Vertices |
| ZERO-SHOT | N-Pair | 18.4% | 99.4% | 41.2% | 93.0% | 91.6% | 60.2% | 51.8% | 34.0% |
| | LO-Inc | 59.2% | 100.0% | 40.4% | 97.4% | 93.8% | 67.4% | 79.4% | 40.8% |
| | Adj-Mat | 57.6% | 100.0% | 23.0% | 92.2% | 73.2% | 47.6% | 19.8% | 14.0% |
| | Avg. | 45.1% | 99.8% | 34.9% | 94.2% | 86.2% | 58.4% | 50.3% | 29.6% |
| | N-Set | 85.8% | 99.6% | 90.4% | 89.6% | 88.8% | 58.6% | 46.2% | 37.8% |
| | HO-Inc | 88.2% | 100.0% | 53.8% | 84.0% | 91.8% | 70.4% | 67.4% | 39.8% |
| | Inc-Mat | 95.8% | 99.8% | 50.6% | 79.8% | 67.8% | 33.8% | 9.4% | 8.6% |
| | HO-Neigh | 90.2% | 100.0% | 85.8% | 87.6% | 93.4% | 56.4% | 33.0% | 25.0% |
| | Avg. | 90.0% | 99.9% | 70.2% | 85.3% | 85.5% | 54.8% | 39.0% | 27.8% |
| ZERO-HYPER-COT | N-Pair | 20.6% | 75.0% | 49.0% | 94.6% | 95.6% | 62.9% | 70.6% | 39.4% |
| | LO-Inc | 53.0% | 99.8% | 53.8% | 99.8% | 98.4% | 69.8% | 75.6% | 57.2% |
| | Adj-Mat | 56.4% | 99.2% | 21.4% | 94.0% | 73.2% | 51.8% | 16.6% | 7.2% |
| | Avg. | 43.3% | 91.3% | 41.4% | 96.1% | 89.1% | 61.5% | 54.3% | 34.6% |
| | N-Set | 96.2% | 54.6% | 99.2% | 91.4% | 91.6% | 63.2% | 73.4% | 46.4% |
| | HO-Inc | 97.2% | 100.0% | 97.4% | 100.0% | 99.2% | 72.0% | 73.8% | 58.6% |
| | Inc-Mat | 76.0% | 99.4% | 67.8% | 79.4% | 69.4% | 33.0% | 16.4% | 1.8% |
| | HO-Neigh | 91.4% | 89.4% | 97.6% | 96.0% | 95.6% | 58.6% | 58.6% | 47.4% |
| | Avg. | 90.2% | 85.9% | 90.5% | 91.7% | 89.0% | 56.7% | 55.6% | 38.6% |
| FEW-SHOT | N-Pair | 65.0% | 99.2% | 49.0% | 93.8% | 91.0% | 74.2% | 65.6% | 30.4% |
| | LO-Inc | 95.6% | 99.0% | 45.4% | 96.0% | 93.6% | 79.4% | 92.8% | 47.2% |
| | Adj-Mat | 79.2% | 100.0% | 33.8% | 87.2% | 85.6% | 64.8% | 34.2% | 23.8% |
| | Avg. | 79.9% | 99.4% | 42.7% | 92.3% | 90.1% | 72.8% | 64.2% | 33.8% |
| | N-Set | 84.0% | 99.6% | 78.0% | 94.0% | 88.8% | 75.2% | 55.8% | 31.0% |
| | HO-Inc | 98.6% | 100.0% | 80.0% | 98.2% | 87.4% | 74.8% | 75.6% | 45.4% |
| | Inc-Mat | 98.8% | 99.8% | 58.0% | 73.0% | 62.2% | 46.9% | 12.0% | 14.8% |
| | HO-Neigh | 97.8% | 98.0% | 96.4% | 90.4% | 88.6% | 63.2% | 40.4% | 23.4% |
| | Avg. | 94.8% | 99.4% | 78.1% | 88.9% | 81.8% | 65.0% | 46.0% | 28.7% |
| COT | N-Pair | 90.6% | 99.6% | 48.0% | 93.8% | 91.6% | 66.4% | 63.2% | 32.2% |
| | LO-Inc | 97.4% | 99.6% | 45.6% | 95.6% | 94.0% | 73.4% | 92.8% | 48.2% |
| | Adj-Mat | 92.6% | 99.8% | 38.8% | 87.2% | 87.6% | 64.6% | 35.4% | 24.2% |
| | Avg. | 93.5% | 99.7% | 44.1% | 92.2% | 91.1% | 68.1% | 63.8% | 34.9% |
| | N-Set | 82.6% | 99.8% | 81.6% | 94.0% | 89.4% | 71.6% | 59.4% | 31.4% |
| | HO-Inc | 97.2% | 100.0% | 87.2% | 97.6% | 87.6% | 74.0% | 78.2% | 44.8% |
| | Inc-Mat | 97.8% | 99.4% | 60.4% | 74.2% | 66.6% | 44.4% | 13.0% | 15.0% |
| | HO-Neigh | 96.6% | 97.6% | 96.4% | 92.2% | 88.6% | 60.4% | 42.8% | 24.0% |
| | Avg. | 93.6% | 99.2% | 81.4% | 89.5% | 83.1% | 62.6% | 48.4% | 28.8% |
| COT-HYPER-BAG | N-Pair | 87.8% | 100.0% | 44.4% | 95.0% | 92.0% | 70.8% | 62.0% | 32.4% |
| | LO-Inc | 95.8% | 99.8% | 47.0% | 96.4% | 94.0% | 78.4% | 90.8% | 46.2% |
| | Adj-Mat | 89.2% | 100.0% | 36.4% | 88.4% | 87.8% | 67.2% | 35.2% | 25.0% |
| | Avg. | 90.9% | 99.9% | 42.6% | 93.3% | 91.3% | 72.1% | 62.7% | 34.5% |
| | N-Set | 93.6% | 100.0% | 79.4% | 93.6% | 86.8% | 74.6% | 55.8% | 29.8% |
| | HO-Inc | 98.4% | 100.0% | 84.8% | 97.8% | 87.0% | 76.8% | 77.2% | 43.2% |
| | Inc-Mat | 97.6% | 99.6% | 60.2% | 77.2% | 72.0% | 47.2% | 11.8% | 16.0% |
| | HO-Neigh | 97.2% | 98.8% | 96.2% | 90.0% | 87.0% | 67.2% | 37.4% | 22.6% |
| | Avg. | 96.7% | 99.6% | 80.2% | 89.7% | 83.2% | 66.5% | 45.6% | 27.9% |
| Overall Avg. | | 83.4% | 97.3% | 63.4% | 91.0% | 86.6% | 63.5% | 52.1% | 31.7% |

Table 3: Results of different prompting frameworks and hypergraph languages.

| | | High-Order Tasks | | | | Isomorphism Tasks | |
| --- | --- | --- | --- | --- | --- | --- | --- |
| Difficulty | Easy | | Medium | | Hard | Hard | |
| Settings | Hyperedge Degree | Vertex Set Connection | VS-in-He Check | He-in-He Check | Shared-Vertices between He | Ismorphism Recognition | Structure Classification |
| **ZERO-SHOT** | | | | | | | |
| N-Pair | 51.6% | 65.4% | 73.0% | 33.4% | 30.3% | 52.0% | 25.2% |
| LO-Inc | 22.6% | 77.4% | 82.2% | 54.6% | 25.1% | 47.0% | 31.9% |
| Adj-Mat | 19.6% | 14.6% | 10.6% | 51.4% | 35.7% | 50.6% | 29.2% |
| Avg. | 31.3% | 52.5% | 55.3% | 46.5% | 30.4% | 49.9% | 28.8% |
| N-Set | 99.6% | 92.6% | 85.8% | 67.0% | 51.7% | 46.2% | 41.2% |
| HO-Inc | 82.0% | 83.8% | 91.2% | 61.6% | 31.3% | 50.6% | 18.2% |
| Inc-Mat | 39.2% | 4.8% | 1.4% | 41.6% | 45.4% | 64.6% | 36.3% |
| HO-Neigh | 98.2% | 82.2% | 74.8% | 57.2% | 32.0% | 51.2% | 31.2% |
| Avg. | 79.8% | 65.9% | 63.3% | 56.9% | 40.1% | 53.2% | 31.7% |
| **ZERO-HYPER-COT** | | | | | | | |
| N-Pair | 54.2% | 64.2% | 79.6% | 33.6% | 29.7% | 53.2% | 27.0% |
| LO-Inc | 21.2% | 83.6% | 83.8% | 51.4% | 28.8% | 51.6% | 31.4% |
| Adj-Mat | 19.2% | 13.0% | 4.6% | 53.2% | 31.7% | 48.6% | 25.7% |
| Avg. | 31.5% | 53.6% | 56.0% | 46.1% | 30.1% | 51.1% | 28.0% |
| N-Set | 52.8% | 93.8% | 92.0% | 60.8% | 91.5% | 47.8% | 32.3% |
| HO-Inc | 97.0% | 96.6% | 98.0% | 52.0% | 96.9% | 52.8% | 31.0% |
| Inc-Mat | 52.0% | 6.4% | 2.6% | 41.0% | 44.4% | 64.8% | 26.1% |
| HO-Neigh | 100.0% | 97.8% | 98.6% | 47.2% | 94.4% | 56.4% | 38.5% |
| Avg. | 75.5% | 73.7% | 72.8% | 50.3% | 81.8% | 55.5% | 32.0% |
| **FEW-SHOT** | | | | | | | |
| N-Pair | 56.4% | 78.6% | 81.2% | 37.2% | 22.6% | 45.0% | 30.5% |
| LO-Inc | 33.2% | 84.4% | 90.8% | 62.2% | 20.7% | 44.9% | 42.0% |
| Adj-Mat | 26.2% | 68.8% | 76.8% | 53.2% | 23.0% | 41.4% | 38.5% |
| Avg. | 38.6% | 77.3% | 82.9% | 50.9% | 22.1% | 43.8% | 37.0% |
| N-Set | 99.6% | 83.4% | 87.2% | 82.6% | 89.2% | 43.3% | 82.7% |
| HO-Inc | 84.8% | 87.8% | 93.6% | 65.2% | 51.7% | 44.2% | 43.8% |
| Inc-Mat | 40.6% | 44.0% | 53.2% | 48.2% | 23.0% | 47.8% | 53.5% |
| HO-Neigh | 99.8% | 81.4% | 87.2% | 65.2% | 62.7% | 47.7% | 65.5% |
| Avg. | 81.2% | 74.2% | 80.3% | 65.3% | 56.7% | 45.8% | 61.4% |
| **COT** | | | | | | | |
| N-Pair | 49.8% | 80.8% | 76.4% | 40.0% | 25.7% | 47.2% | 28.3% |
| LO-Inc | 34.0% | 84.4% | 82.4% | 59.0% | 18.5% | 46.6% | 41.8% |
| Adj-Mat | 25.4% | 74.6% | 77.4% | 62.4% | 29.3% | 44.2% | 40.3% |
| Avg. | 36.4% | 79.9% | 78.7% | 53.8% | 24.5% | 46.0% | 36.8% |
| N-Set | 100.0% | 84.0% | 87.2% | 77.4% | 91.1% | 47.2% | 82.3% |
| HO-Inc | 86.6% | 88.4% | 88.4% | 69.8% | 55.2% | 46.0% | 41.6% |
| Inc-Mat | 42.6% | 47.8% | 53.6% | 57.4% | 24.1% | 60.0% | 52.2% |
| HO-Neigh | 100.0% | 84.2% | 82.6% | 66.6% | 62.0% | 49.2% | 66.4% |
| Avg. | 82.3% | 76.1% | 78.0% | 67.8% | 58.1% | 50.6% | 60.6% |
| **COT-HYPER-BAG** | | | | | | | |
| N-Pair | 53.2% | 77.4% | 75.0% | 38.8% | 24.9% | 47.0% | 27.4% |
| LO-Inc | 32.8% | 86.4% | 82.2% | 65.4% | 23.9% | 46.5% | 39.4% |
| Adj-Mat | 24.0% | 49.8% | 66.4% | 64.6% | 33.6% | 47.4% | 38.9% |
| Avg. | 36.7% | 71.2% | 74.5% | 56.3% | 27.5% | 47.0% | 35.2% |
| N-Set | 100.0% | 84.6% | 87.6% | 81.4% | 89.4% | 46.6% | 87.2% |
| HO-Inc | 86.6% | 90.4% | 88.2% | 77.2% | 53.5% | 48.2% | 38.9% |
| Inc-Mat | 44.6% | 27.2% | 43.0% | 58.2% | 28.4% | 58.8% | 49.6% |
| HO-Neigh | 100.0% | 83.0% | 81.6% | 70.4% | 61.8% | 56.6% | 65.0% |
| Avg. | 82.8% | 71.3% | 75.1% | 71.8% | 58.3% | 52.6% | 60.2% |
| Overall Avg. | 60.8% | 69.9% | 72.0% | 57.4% | 45.2% | 49.8% | 42.3% |

**Observation 2: High-order structure languages enhance LLMs' comprehension of vertex set correlations within hypergraphs.** As shown in Table 3, high-order structure languages significantly outperform low-order structure languages in high-order and isomorphism tasks across all difficulty levels. This improvement stems from high-order tasks focusing on vertex set correlations, which low-order languages find challenging to describe. High-order languages effectively represent complex multi-vertex structures, enabling LLMs to perform accurate computations and provide precise answers for intricate relationships. Consequently, using high-order structure languages significantly boosts LLMs' understanding of vertex set correlations within hypergraphs.

**Observation 3: Current LLMs struggle with isomorphism recognition tasks, but example-based prompting can partially bridge this gap.** In binary classification isomorphism tasks, LLMs achieve around 50% accuracy (Table 3), showing that existing hypergraph textualizations are inad-

equate. In three-class structure classification, low-order structure languages produce approximately 30% accuracy, failing to distinguish macro structures, and high-order structure languages alone offer little improvement. However, combining high-order structure languages with example-based prompting increases accuracy to over 60%, suggesting that providing examples can aid LLMs in better understanding high-order structures and distinguishing between different macro-structures.

**Observation 4: In-context learning methods enhance LLMs' understanding of hypergraph structures and task logic, thereby improving accuracy.** Our experiments in Table 2 and Table 3 show that providing examples, such as Few-Shot and CoT prompts, significantly boost LLM performance compared to scenarios without examples. These examples help LLMs grasp the task logic, follow instructions more effectively, and produce accurate results. Specifically, settings with examples exhibit notable accuracy increases, highlighting the crucial role of in-context learning in understanding complex hypergraph structures and task requirements.

In the following, we selected two typical tasks from each of the three hypergraph categories to study the LLMs' understanding of hypergraphs in depth.

## 4.2 RESULTS OF DIFFERENT LLMS

We evaluate six mainstream LLMs on hypergraph tasks, as shown in Figure 1. Larger models like GPT-4 outperform smaller ones such as GPT-3.5, highlighting the importance of model capacity in understanding complex hypergraph relationships. While LLMs handle low-order and high-order tasks—analyzing local and multi-vertex associations—they struggle with isomorphism recognition, which requires a comprehensive understanding of entire hypergraph structures beyond their current visualization and global pattern recognition abilities.

**Observation 5: Enhancing LLM capabilities improves hypergraph understanding, but current models generally cannot solve isomorphism recognition tasks for macro-structures.** Our evaluation shows that increasing model power aids hypergraph comprehension, yet existing LLMs fail to recognize and distinguish complex global structures within hypergraphs. This limitation underscores the need for future research to enhance LLMs' ability to visualize and internally represent high-order relational dependencies essential for effective isomorphism recognition. Consequently, only low-order and high-order tasks are used for further evaluation.

## 4.3 RESULTS OF DIFFERENT HYPER-COT PROMPTINGS

**Observation 6: Hyper-COT enhances LLMs' hypergraph comprehension.** Hypergraphs' high-order correlations challenge LLMs more than low-order structures. We introduce *Hyper-COT*, adding step-by-step prompts. Tested on structure classification tasks (Table 4), Hyper-COT consistently outperforms the "Naive COT". Note that "Naive COT" refers to "Let's think step by step". Specifically, Hyper-COT v3 boosts performance by 4% on average across seven encodings and up to 9% in N-Pair and N-Set encodings by guiding LLMs to interpret connections hierarchically.

Table 4: Resutls of different Hyper-COT prompts on the structure classification task.

| | Low-Order Language | | | High-Order Language | | | | Avg. |
| | N-Pair | LO-Inc | Adj-Mat | N-Set | HO-Inc | Inc-Mat | HO-Neigh | Rank |
|---|---|---|---|---|---|---|---|---|
| Naive COT | 24.3% | 30.1% | 27.0% | 27.0% | 25.2% | 23.0% | 32.3% | 2.86 |
| Hyper-COT v1 | 22.1% | 27.4% | 26.1% | 29.6% | 25.8% | 23.9% | 31.4% | 2.86 |
| Hyper-COT v2 | 22.1% | 23.5% | 27.1% | 28.8% | 27.0% | 23.5% | 31.9% | 2.71 |
| Hyper-COT v3 | 33.6% | 36.7% | 27.9% | 36.0% | 20.9% | 29.2% | 32.4% | 1.43 |

## 4.4 RESULTS OF HYPER-BAG PROMPTING

**Observation 7: Hyper-BAG enhances LLMs' performance on high-order tasks.** While Build-A-Graph (BAG) prompting Wang et al. (2024) is effective for graph understanding, we introduce *Hyper-BAG* tailored for hypergraphs. Experiments in Table 5 show that BAG does not significantly improve simple low-order tasks. However, both BAG and Hyper-BAG enhance performance by over 2% in high-order tasks, with Hyper-BAG achieving a 2.8% improvement in the Vertex Set Connection task compared to no BAG usage. These findings demonstrate that Hyper-BAG effectively aids LLMs in understanding and performing high-order hypergraph tasks.

Table 5: The influence of using additional Hyper-BAG prompting under different settings.

| Settings | Low-Order Tasks | | High-Order Tasks | |
|---|---|---|---|---|
| | Vertex Con. | Rea. Check | VS Connection | VS-in-He Check |
| w.o. | 94.0% | 88.8% | 83.4% | 87.2% |
| BAG | 93.0% | 90.2% | 85.2% | 89.2% |
| COT-BAG | 93.6% | 87.8% | 84.6% | 87.2% |
| Hyper-BAG | 93.6% | 89.4% | 86.2% | 89.8% |

Table 6: The influence of the Hyper-COT with k-shot step-by-step demonstration.

| #Shot | Low-Order Tasks | | High-Order Tasks | |
|---|---|---|---|---|
| | Vertex. Con. | Rea. Check | VS Connection | VS-in-He Check |
| 0 | 83.8% | 91.4% | 84.4% | 84.4% |
| 1 | 82.4% | 79.2% | 90.2% | 86.4% |
| 2 | 96.2% | 84.4% | 87.8% | 93.4% |
| 3 | 97.8% | 85.8% | 90.4% | 92.6% |

## 4.5 RESULTS OF HYPER-COT WITH DIFFERENT NUMBER OF SHOTS

**Observation 8: Providing example solutions enhances LLMs' understanding of hypergraphs.**
In structure classification tasks, supplying example solutions improves LLMs' comprehension of hypergraphs. Experiments in Table 6 show that increasing the number of examples does not enhance performance in low-order tasks. However, in high-order tasks, more examples lead to significant performance gains of approximately 6% to 9% compared to no examples. Specifically, example prompts help LLMs better understand and distinguish complex hypergraph structures, thereby enabling more accurate classifications.

## 4.6 RESULTS OF DIFFERENT HYPERGRAPH SIZE ON REAL-WORLD HYPERGRAPHS

**Observation 9: Larger hypergraphs pose greater challenges for LLMs' comprehension.** We further investigated the impact of hypergraph size on the ability of LLMs to understand hypergraph structures. The experimental results are presented in Table 7. Considering that real-world hypergraphs exhibit certain regularities as the number of vertices increases, we sampled hypergraph structures from citation networks and protein structures for this experiment. The hypergraph sampling method and size settings are detailed in Appendix D. From the results, we observe that as the size of the hypergraph increases, the performance of LLMs deteriorates. This trend is more pronounced in high-order tasks compared to low-order tasks. Specifically, in citation data, the performance on high-order tasks declines by up to 13% as the hypergraph size grows. This degradation is attributed to the increased complexity of larger hypergraph structures, which hinders LLMs' ability to comprehend and reason effectively.

Table 7: Influence of hypergraph size on real-world citation and protein datasets.

| | | Low-Order Tasks | | High-Order Tasks | |
|---|---|---|---|---|---|
| | | Vertex Connection | Reachability Check | Vertex Set Connection | Vertex-Set-in-Hyperedge Check |
| Citation | Small | 96.0% | 90.0% | 95.0% | 94.0% |
| | Middle | 100.0% | 82.0% | 93.0% | 90.0% |
| | Large | 97.0% | 77.0% | 91.0% | 81.0% |
| Protein | Small | 86.0% | 92.0% | 96.0% | 92.0% |
| | Middle | 90.0% | 91.0% | 94.5% | 86.0% |
| | Large | 90.0% | 88.0% | 94.0% | 88.0% |

## 5 CONCLUSION

In this paper, we introduce `LLM4Hypergraph`, the first benchmark designed to evaluate and enhance LLMs' understanding of hypergraph-structured data. Our benchmark includes 21,500 tasks covering various low-order, high-order, and isomorphism challenges using both synthetic and real-world hypergraphs. Evaluating six prominent LLMs, we demonstrate the benchmark's ability to identify model strengths and weaknesses. Additionally, our specialized prompting framework, featuring *Hyper-BAG* and *Hyper-COT*, boosts reasoning performance by an average 4% (up to 9%) on the structure classification task. This work lays the groundwork for future integration of hypergraph computational capabilities into LLMs, fostering more advanced data comprehension.

ACKNOWLEDGMENTS

This work was supported by the National Natural Science Foundation of China under Grant (No. U24A20252, 623B2066), the National Major Scientific Instruments and Equipments Development Project of National Natural Science Foundation of China under Grant 62327808, Beijing Natural Science Foundation (No. L242167), the Key Research and Development Program of Shaanxi Province of China under Grant Nos. 2024PT-ZCK-66 and 2024CY2-GJHX-48, Guangdong Major Project of Basic and Applied Basic Research under Grant 2023B0303000009, National Key Laboratory of Human-Machine Hybrid Augmented Intelligence, Xi'an Jiaotong University (No. HMHAI-202412). We would like to express our sincere gratitude to OPPO for their generous support of our work.

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

## A  RELATED WORKS

**LLMs for Graphs.** LLMs' ability to address graph-based problems has garnered significant attention. Benchmarks like `NLGraph` Wang et al. (2024), `GraphInstruct` Luo et al. (2024), `GraphQA` Fatemi et al. (2023), and `LLM4DyG` Zhang et al. (2023) demonstrate LLMs' potential in handling structured and dynamic graph data, emphasizing the importance of encoding strategies. Additionally, in structured commonsense reasoning, LLMs generate various graph structures from natural language inputs (Tandon et al., 2019; Madaan et al., 2022; Saha et al., 2021). However, LLMs' ability to perform reasoning within hypergraphs remains underexplored. To bridge this gap, we introduce the benchmark, designed to enhance LLMs' understanding of hypergraph tasks.

**LLMs for Few-Shot Reasoning.** Extensive research has assessed LLMs' few-shot reasoning across arithmetic, logical, and commonsense domains. Arithmetic tasks use datasets like AQUA (Ling et al., 2017), GSM8K (Cobbe et al., 2021), and SVAMP (Patel et al., 2021). For mathematical reasoning, NaturalProofs (Welleck et al., 2022) evaluates LLMs' ability to generate proof steps and complete proofs. Logical reasoning includes symbolic problems such as Coin Flip and Last Letter Concatenation (Wei et al., 2022), and the Logic Grid Puzzle from BIG-BENCH (Srivastava et al., 2022). Commonsense reasoning utilizes datasets from Talmor et al. (2018) and Geva et al. (2021). Additionally, methods to enhance and evaluate LLMs' algorithmic reasoning have been developed (Zhou et al., 2022b; Lee & Kim, 2023; Liu et al., 2023). Despite these efforts, achieving reliable and deep reasoning remains challenging. To address this, we introduce the `LLM4Hypergraph`, aimed at improving the assessment of LLMs' reasoning abilities in hypergraph-based tasks.

## B  EXPERIMENTS OF THE UNCERTAINTY QUANTIFICATION

To comprehensively assess the capability of LLMs in comprehending hypergraphs, we designed a series of experiments focusing on their performance across various hypergraph-related tasks. Recognizing the importance of uncertainty quantification and error characterization, we introduced controlled randomness by setting the temperature parameter to 0.8. This setup allows for the examination of model stability and consistency in output generation. We evaluated six prominent LLMs: ERNIE-Lite-8K, ERNIE-Speed-128K, Qwen-Long, LLaMA3-8B, GPT-3.5 Turbo, and GPT-4o. For each model, five independent runs were conducted to ensure the reliability of the results.

The evaluation encompassed six distinct tasks categorized into three types: low-order tasks (Vertex Connection Check and Reachability Check), high-order tasks (Vertex Set Connection Check and Vertex-Set-in-Hyperedge Check), and isomorphism tasks (Isomorphism Recognition and Structure Classification). Table 8 presents the mean accuracy and standard error for each model across these tasks.

Table 8: Mean accuracy (%) and standard error for each model across various tasks.

| Model | Low-Order Tasks | | High-Order Tasks | | Isomorphism Tasks | |
|---|---|---|---|---|---|---|
| | Vertex Connection | Reachability Check | Vertex Set Connection | VS-in-He Check | Isomorphism Recognition | Structure Classification |
| ERNIE-Lite-8K | $82.12_{\pm 0.43}$ | $78.28_{\pm 1.49}$ | $67.52_{\pm 0.83}$ | $77.52_{\pm 2.13}$ | $43.24_{\pm 0.14}$ | $42.58_{\pm 4.69}$ |
| ERNIE-Speed-128K | $78.64_{\pm 0.06}$ | $97.56_{\pm 0.02}$ | $67.20_{\pm 0.22}$ | $71.24_{\pm 2.60}$ | $43.84_{\pm 0.14}$ | $22.83_{\pm 0.20}$ |
| Qwen-Long | $97.56_{\pm 0.16}$ | $98.24_{\pm 0.24}$ | $73.96_{\pm 0.28}$ | $88.68_{\pm 0.13}$ | $44.60_{\pm 0.02}$ | $44.94_{\pm 0.05}$ |
| LLaMA3-8B | $79.48_{\pm 1.35}$ | $82.60_{\pm 6.34}$ | $71.80_{\pm 3.50}$ | $78.40_{\pm 4.18}$ | $47.72_{\pm 1.25}$ | $23.70_{\pm 4.47}$ |
| GPT-3.5 Turbo | $73.68_{\pm 1.01}$ | $74.12_{\pm 1.29}$ | $58.64_{\pm 1.34}$ | $70.40_{\pm 1.10}$ | $44.68_{\pm 0.01}$ | $27.60_{\pm 3.48}$ |
| GPT-4o | $66.68_{\pm 0.05}$ | $99.48_{\pm 0.03}$ | $96.36_{\pm 0.06}$ | $98.68_{\pm 0.13}$ | $44.04_{\pm 0.10}$ | $27.32_{\pm 8.68}$ |

The results reveal several key insights into the performance and reliability of the evaluated LLMs. Firstly, most models exhibit low standard errors across various tasks, indicating high consistency and reliability in their performance. For example, Qwen-Long achieved a mean accuracy of $97.56 \pm 0.16\%$ on the Vertex Connection Check task, demonstrating exceptional stability. Secondly, significant variability exists in the performance of different models on the same tasks. While Qwen-Long excels in the Reachability Check task with an accuracy of $98.24 \pm 0.24\%$, ERNIE-Speed-128K underperforms in the Structure Classification task, attaining only $22.83 \pm 0.20\%$. This variability highlights inherent differences in the capabilities of each model rather than being attributable to random errors.

Furthermore, high-order tasks present greater challenges, as evidenced by higher standard errors in some models. Notably, GPT-4o exhibits a substantial standard error of 8.68% in the Structure Classification task, underscoring the increased complexity involved in high-order relational reasoning. This indicates that while LLMs can handle low-order relationships with high consistency, their performance in more complex, high-order tasks remains variable and warrants further investigation.

In summary, the comprehensive experimental evaluation demonstrates that the evaluated LLMs maintain consistent and reliable performance across a range of hypergraph-related tasks, with minimal impact from random errors. However, significant performance differences among models and challenges in high-order tasks highlight the distinct strengths and limitations of each architecture. These findings provide a robust foundation for future research aimed at enhancing the hypergraph comprehension abilities of LLMs.

## C    IMPACT OF HYPERGRAPH DOMAINS ON LLM PERFORMANCE

To explore how the performance of LLMs varies across different hypergraph domains, we conducted an additional set of experiments using real-world hypergraph datasets. The motivation behind this investigation stems from the recognition that hypergraph structures can significantly influence the ability of LLMs to comprehend and reason about complex relationships. Specifically, we aimed to assess both low-order and high-order understanding capabilities of LLMs within varying hypergraph contexts.

We selected two representative datasets: the **Coauthorship dataset** and the **Protein dataset**. These were chosen based on their distinct average hyperedge degrees, with the Coauthorship dataset exhibiting an average hyperedge degree of 3.34 and the Protein dataset an average of 2.75, compared to traditional low-order structured graphs which typically have an average hyperedge degree of 2. This distinction allowed us to quantify the high-order nature of each dataset and evaluate the models' performance accordingly. The experiments focused on two primary tasks: the **Vertex Connection Check Task** as a representative low-order node understanding task, and the **Vertex-Set-in-Hyperedge Check Task** as a representative high-order hyperedge understanding task. To encode the hypergraph structures, we employed four distinct high-order encoding methods—N-Set, HO-Inc, Inc-Mat, and HO-Neigh—ensuring a comprehensive evaluation of different encoding strategies.

Table 9: Mean accuracy (%) for each model across different hypergraph Domains and Tasks

| Model | Low-Order Task | | High-Order Task | |
|---|---|---|---|---|
| | Coauthorship | Protein | Coauthorship | Protein |
| Averaged Hyperedge Degree | 3.34 | 2.75 | 3.34 | 2.75 |
| N-Set | 98.0 | 97.6 | 95.6 | 94.6 |
| HO-Inc | 99.6 | 99.6 | 97.2 | 87.2 |
| Inc-Mat | 79.8 | 72.8 | 91.2 | 94.6 |
| HO-Neigh | 76.4 | 87.6 | 88.4 | 81.6 |
| **Averaged Results** | **88.4** | **89.4** | **93.1** | **89.5** |

Table 9 presents the mean accuracy for each model across the different tasks and datasets. The results reveal that for the high-order **Vertex-Set-in-Hyperedge Check Task**, the performance of LLMs improved as the hyperedge degree of the dataset increased, with an approximate enhancement of 3.6%. This suggests that the high-order encoding methods effectively enhance the models' ability to handle complex relational tasks, particularly benefiting from the richer high-order structures present in the Coauthorship dataset. In contrast, for the low-order **Vertex Connection Check Task**, a lower hyperedge degree correlated with better performance. Datasets with hyperedge degrees closer to traditional graph structures facilitated higher accuracy, likely due to reduced redundancy in high-order descriptive language, which simplifies structural comprehension for LLMs.

Furthermore, our findings confirm that variations in hypergraph domains lead to significant differences in LLM performance. These differences are fundamentally attributable to variations in hypergraph data distributions across domains, such as the sparsity of connections and the density of hyperedges. For instance, the Protein dataset, with a lower average hyperedge degree of 2.75,

supported better performance in low-order tasks compared to the Coauthorship dataset. This underscores the importance of considering domain-specific characteristics when evaluating and designing LLMs for hypergraph comprehension.

In summary, the extended experiments demonstrate that the performance of LLMs is inherently influenced by the characteristics of hypergraph domains. Higher-order structures in datasets like Coauthorship enhance LLMs' capabilities in complex relational tasks, while lower-order structures in datasets such as Protein support better performance in simpler tasks. These insights highlight the necessity of tailoring hypergraph encoding strategies to specific domain characteristics to optimize LLM performance. Future work will involve expanding this analysis to additional real-world hypergraph datasets from diverse domains to further elucidate the domain-specific impacts on LLM performance.

## D  HYPERGRAPH GENERATION

We create a diverse series of hypergraphs that cover both synthetic and real-world instances. The `LLM4Hypergraph` Benchmark encompasses a wide spectrum of hypergraph types and scales, thereby providing a robust and versatile framework for assessing the proficiency of large language models in understanding and processing hypergraph-based data structures.

### D.1  SYNTHETIC HYPERGRAPHS

We first introduce the construction of synthetic hypergraphs. The synthetic hypergraphs can be categorized into two distinct types: random hypergraphs and regular-structured hypergraphs. Random hypergraphs are generated using the 'hypergraph_Gnm()' function[*] from the DHG toolkit[*]. Importantly, we configure these hypergraphs with a "low-order first" structure. This configuration is grounded in the principle of Occam's razor, which posits that simpler, lower-order associations are more prevalent and thus more representative of real-world scenarios where low-order relationships typically dominate over high-order ones. Consequently, random hypergraphs are extensively utilized across various tasks within the benchmark, with the exception of isomorphic tasks where structured classification is paramount. In contrast, regular-structured hypergraphs are specifically tailored for structural classification tasks. We select three classical hypergraph structures Feng et al. (2024) known for their distinct and regular patterns: the Hyper Pyramid, Hyper Checked Table, and Hyper Wheel. To introduce variability and enhance the robustness of the benchmark, we systematically modify the hyperparameters of these base structures. For instance, we vary the number of layers in the Hyper Pyramid and the number of vertices per blade in the Hyper Wheel, thereby generating a multitude of structurally diverse yet regularly patterned hypergraphs.

### D.2  REAL-WORLD HYPERGRAPHS

As for the real-world instances, we source data from citation networks and protein structure datasets. The construction process involves randomly sampling sub-hypergraphs by selecting a central vertex and performing a random walk through selected hyperedges. During this walk, the vertices encountered are randomly retained until the sub-hypergraph reaches the predetermined vertex count. This methodology ensures that the sampled hypergraphs retain the intricate and authentic correlations inherent in real-world data.

### D.3  MULTI-SCALE SETTINGS

To ensure scalability and applicability across a range of scenarios, both synthetic and real-world hypergraphs are categorized based on their size, defined by the number of vertices: small-scale hypergraphs (Containing between 5 to 10 vertices), medium-scale hypergraphs (Containing between 10 to 15 vertices), and large-scale hypergraphs (containing between 15 to 20 vertices). Additionally, to maintain structural generality, synthetic hypergraphs adhere to a hyperedge-to-vertex ratio ranging from 0.2 to 1.5 times the number of vertices. This ratio ensures a balanced complexity that is neither

---

[*]dhg.random.hypergraph_Gnm()
[*]www.deephypergraph.com

too sparse nor excessively dense, thereby facilitating meaningful evaluations of LLM capabilities. In contrast, real hypergraphs do not impose such restrictions on the number of hyperedges, allowing them to naturally reflect the inherent connectivity patterns of their source datasets.

# E    DETAILS OF TASK DESIGN

In this section, we will introduce the proposed three types of tasks in detail, respectively.

## E.1    ISOMORPHISM TASKS

In the LLM4Hypergraph Benchmark, we prioritize isomorphic tasks as a fundamental component of our evaluation framework. This prioritization is grounded in the belief that a comprehensive understanding of the overall structural architecture is essential for LLMs to be effectively utilized in practical applications. Isomorphic tasks serve as a theoretical cornerstone for numerous model capabilities, particularly in specialized fields such as molecular structure analysis and protein analysis. In these domains, proteins with similar global structures but differing local configurations can exhibit entirely distinct properties. This phenomenon underscores the necessity for models to possess a heightened sensitivity to structural isomorphism, enabling them to accurately recognize and differentiate between nuanced structural variations. To evaluate this capability, we introduce two representative isomorphic tasks within our benchmark:

- *Isomorphism Recognition.* This task assesses the model's ability to determine whether two different representations accurately reflect the same underlying hypergraph structure. Given two hypergraphs $\mathcal{G} = \{\mathcal{V}, \mathcal{E}\}$ and $\mathcal{G}' = \{\mathcal{V}', \mathcal{E}'\}$, the target is to find whether a bijective mapping $g := \mathcal{V} \rightarrow \mathcal{V}'$ exists. The mapping $g$ is called the isomorphism function, such that $(v_1, v_2, \ldots, v_m) \in \mathcal{E} \iff (g(v_1), g(v_2), \ldots, g(v_m)) \in \mathcal{E}'$. Note that if $|\mathcal{V}| \neq |\mathcal{V}'|$, we can directly have $\mathcal{G}$ and $\mathcal{G}'$ are not isomorphism.

- *Structure Classification.* This task evaluates the model's proficiency in distinguishing hypergraphs based on their macro-level architectural frameworks. It requires the model to identify and understand differences in the overall structural layout, even when local configurations may appear similar. Let $\mathbb{S}$ denote an equivalence class set of hypergraphs, encompassing several distinct types of hypergraphs $\{\mathcal{T}_1, \mathcal{T}_2, \ldots, \mathcal{T}_n\}$. Each subtype $\mathcal{T}_i$ within $\mathbb{S}$ adheres to a specific topological rule or structural pattern that defines its global architecture. These topological rules encapsulate the macroscopic organizational principles that distinguish one hypergraph type from another within the equivalence class. Prior to the task, the LLMs is provided with the fundamental topological rules governing each hypergraph subtype in $\mathcal{T}$. The model is expected to internalize these structural paradigms to facilitate accurate classification. This task emphasizes the model's ability to comprehend and differentiate hypergraphs based on their overarching structural frameworks, rather than merely local or pairwise relationships. Accurate performance on the Macroscopic Architecture Differentiation task indicates that the LLM possesses a nuanced understanding of hypergraph topology, which is critical for applications in domains such as molecular structure analysis and protein modeling.

## E.2    LOW-ORDER TASKS

In addition to isomorphic tasks, our LLM4Hypergraph Benchmark incorporates a series of low-order tasks designed to enhance LLMs' understanding of fundamental hypergraph properties and the relationships between vertices. These tasks facilitate the comprehension of basic hypergraph attributes and simple connectivity patterns, which are essential for more complex structural analyses.

- *Hyperdge Count.* This task aims to evaluate the model's ability to determine the total number of hyperedges within a given hypergraph. Given a hypergraph $\mathcal{G} = \{\mathcal{V}, \mathcal{E}\}$, the target is to return $|\mathcal{E}|$, the cardinality of the hyperedge set $\mathcal{E}$.

- *Vertex Count.* This task assesses the model's capability to ascertain the total number of vertices in a hypergraph. Given $\mathcal{G} = \{\mathcal{V}, \mathcal{E}\}$, the target is to return $|\mathcal{V}|$, the cardinality of the vertex set $\mathcal{V}$.

- *Vertex Degree.* This task assesses the model's capability to count the number of hyperedges that connects a specific vertex. Given $\mathcal{G} = \{\mathcal{V}, \mathcal{E}\}$ and vertex $v$, the target is to return $|\{e \mid v \in e\}|$.

- *Vertex Connection.* This task is designed to evaluate whether a specific pair of vertices is directly connected by at least one hyperedge. Given $\mathcal{G} = \{\mathcal{V}, \mathcal{E}\}$ and a pair of vertices $(u, v)$, the target is to determine if there exists a hyperedge $e \in \mathcal{E}$ such that $u \in e$ and $v \in e$.

- *Connected Vertices.* This task examines, given a vertex $u$, the model's ability to output the set of vertices directly connected to $u$ by at least one hyperedge. Formally, given $\mathcal{G} = \{\mathcal{V}, \mathcal{E}\}$ and a vertex $u \in \mathcal{V}$, the target is to output the subset $\mathcal{W} \subseteq \mathcal{V}$ such that for each $w \in \mathcal{W}$, there exists a hyperedge $e \in \mathcal{E}$ where $u \in e$ and $w \in e$.

- *Disconnected Vertices.* This task requires, given a vertex $u$, the identification and output of the set of vertices that are not connected to $u$ by any hyperedge. Formally, given $\mathcal{G} = \{\mathcal{V}, \mathcal{E}\}$ and a vertex $u \in \mathcal{V}$, the target is to output the subset $\mathcal{Z} \subseteq \mathcal{V}$ where for each $z \in \mathcal{Z}$, there does not exist any hyperedge $e \in \mathcal{E}$ such that $u \in e$ and $z \in e$.

- *Shortest Path.* This task aims to determine, given two vertices $u$ and $v$, the minimal sequence of vertices representing the shortest path connecting them. Formally, given $\mathcal{G} = \{\mathcal{V}, \mathcal{E}\}$ and a pair of vertices $(u, v)$, the target is to find the shortest sequence $[v_1, v_2, \ldots, v_n]$ such that $u = v_1$, $v = v_n$, and for each consecutive pair $(v_i, v_{i+1})$, there exists a hyperedge $e \in \mathcal{E}$ containing both $v_i$ and $v_{i+1}$.

- *Reachability Check.* This task assesses whether one vertex can be reached from another through a sequence of hyperedges. Formally, given $\mathcal{G} = \{\mathcal{V}, \mathcal{E}\}$ and a pair $(u, v)$, the target is to determine if there exists a sequence of hyperedges $e_1, e_2, \ldots, e_k \in \mathcal{E}$ that connects $u$ to $v$.

These low-order tasks collectively ensure that LLMs develop a robust understanding of basic hypergraph properties and simple relational structures. By accurately performing tasks such as counting hyperedges and vertices, determining direct and indirect connections, and assessing reachability, models build a solid foundation for tackling more complex hypergraph-related challenges. The inclusion of these tasks within the LLM4Hypergraph Benchmark not only provides a comprehensive assessment of a model's foundational capabilities but also facilitates the identification of specific areas requiring further enhancement, thereby contributing to the advancement of hypergraph computational intelligence.

### E.3 HIGH-ORDER TASKS

Building upon the foundation of low-order tasks, our LLM4Hypergraph Benchmark introduces a series of high-order tasks tailored to assess LLMs' comprehension of complex correlations between vertex sets within hypergraphs. Unlike low-order tasks that focus on individual vertices and hyperedges, high-order tasks evaluate the model's ability to understand and manipulate relationships involving groups of vertices and their interactions through hyperedges.

- *Hyperedge Degree.* This task aims to determine the degree of a given hyperedge, which is defined as the number of vertices it encompasses. Formally, given a hypergraph $\mathcal{G} = \{\mathcal{V}, \mathcal{E}\}$ and a hyperedge $e \in \mathcal{E}$, the target is to compute $\deg(e) = |\{v \in \mathcal{V} \mid v \in e\}|$.

- *Vertex Set Connection.* This task evaluates whether two distinct sets of vertices $\mathcal{A}$ and $\mathcal{B} \subseteq \mathcal{V}$ are jointly contained within at least one hyperedge, thereby determining if $\mathcal{A}$ and $\mathcal{B}$ are connected through a common hyperedge. Formally, given $\mathcal{G} = \{\mathcal{V}, \mathcal{E}\}$ and two vertex subsets $\mathcal{A}, \mathcal{B} \subseteq \mathcal{V}$, the target is to ascertain whether there exists a hyperedge $e \in \mathcal{E}$ such that $\mathcal{A} \cup \mathcal{B} \subseteq e$.

- *Vertex-Set-in-Hyperedge Check.* This task involves determining whether a specific subset of vertices $\mathcal{S} \subseteq \mathcal{V}$ is entirely contained within any hyperedge of the hypergraph. Formally, given $\mathcal{G} = \{\mathcal{V}, \mathcal{E}\}$ and a vertex subset $\mathcal{S} \subseteq \mathcal{V}$, the target is to evaluate if there exists a hyperedge $e \in \mathcal{E}$ such that $\mathcal{S} \subseteq e$.

- *Hyperedge-in-Hyperedge Check.* This task assesses whether one hyperedge is completely contained within another, thereby identifying hierarchical or nested structures within the

hypergraph. Formally, given $\mathcal{G} = \{\mathcal{V}, \mathcal{E}\}$ and two hyperedges $e_1, e_2 \in \mathcal{E}$, the target is to determine if $e_1 \subseteq e_2$.

- *Shared-Vertices between Hyperedges.* This task requires the model to identify and output the set of vertices shared between any two given hyperedges, thus revealing the overlapping structures and potential intersections within the hypergraph. Formally, given $\mathcal{G} = \{\mathcal{V}, \mathcal{E}\}$ and two hyperedges $e_1, e_2 \in \mathcal{E}$, the target is to return $\mathcal{S} = e_1 \cap e_2$, where $\mathcal{S} \subseteq \mathcal{V}$.

Collectively, these high-order tasks compel LLMs to engage in more sophisticated relational reasoning and structural analysis, enabling a deeper and more comprehensive understanding of hypergraph architectures. By mastering these tasks, models can better handle complex scenarios encountered in real-world applications such as molecular chemistry, social network analysis, and biological systems, where understanding intricate multi-way relationships is paramount. The inclusion of high-order tasks within the LLM4Hypergraph Benchmark thus ensures a thorough evaluation of an LLM's ability to navigate and interpret the multifaceted connectivity inherent in hypergraph data structures, promoting advancements in hypergraph computational intelligence.

### E.4 COMPARISON WITH OTHER BENCHMARKS

In this subsection, we compare our `LLM4Hypergraph` Benchmark with existing graph benchmarks, categorizing them into two main types: purely synthetic graph structure benchmarks and real-world graph structure benchmarks. Prominent examples of purely synthetic benchmarks include NLGraph Wang et al. (2024) and CLEAR Chen et al. (2024), while real-world benchmarks encompass GPT4Graph Guo et al. (2023) and GraphArena Tang et al. (2024). These existing benchmarks predominantly focus on tasks that assess the understanding of local, low-order structural properties within graphs. In contrast, our LLM4Hypergraph Benchmark not only includes low-order structural understanding tasks but also introduces high-order tasks specifically designed to evaluate models' comprehension of associations between sets of vertices. This dual focus enables a more comprehensive exploration of LLMs' ability to grasp complex multi-way relationships inherent in hypergraphs. Moreover, our benchmark uniquely incorporates isomorphic tasks, marking the first instance of such an inclusion in this domain. Isomorphic tasks serve to assess the models' proficiency in understanding and discerning global structural properties of graphs and hypergraphs, thereby providing a more holistic evaluation of their structural comprehension capabilities. Additionally, our benchmark integrates both synthetic hypergraph data and real-world hypergraph data, ensuring a diverse and representative dataset that mirrors the complexities found in practices.

By encompassing both low-order and high-order tasks, along with the novel isomorphic tasks, our `LLM4Hypergraph` Benchmark extends beyond the scope of existing benchmarks. It not only evaluates foundational structural understanding but also delves into advanced relational reasoning and global structural interpretation. This comprehensive approach facilitates a thorough assessment of LLMs' abilities to comprehend and process hypergraph structures, thereby advancing the field of hypergraph computational intelligence. Consequently, our benchmark provides a robust framework for identifying both the strengths and limitations of current models, guiding future improvements in model architecture and training methodologies.

## F DETAILS OF PROMPT FRAMEWORK

The overall prompt framework tailored for hypergraph structures is illustrated in Figure 3. A complete prompt comprises six components: *Example*, *Hypergraph Description*, *Hyper-BAG*, *Question*, *Tail*, and *Hyper-COT*. The *Example* section includes question-answer (QA) pairs for each hypergraph and may additionally contain *COT-for-Answer*, which provides a standard chain-of-thought (CoT) explanation detailing the step-by-step reasoning process to derive the answer. In the depicted figure, certain modules within the prompt are enclosed in brackets "[]", indicating their optional nature depending on the specific prompt configuration employed. We introduce a set of distinct prompt configurations as follows:

- *ZERO-SHOT*: This configuration includes only the *Hypergraph Description*, *Question*, and *Tail* sections, providing the model with the essential information required to address the query without any illustrative examples or additional reasoning prompts.

- *ZERO-HYPER-COT*: Building upon the Zero-Shot setup, this configuration incorporates the *Hyper-COT* component after the *Tail*, introducing a hypergraph-specific chain-of-thought prompt that facilitates structured reasoning tailored to hypergraph-related tasks.

- *FEW-SHOT*: Extending the Zero-Shot approach, the Few-Shot configuration precedes the main prompt with *Example* content. These examples consist solely of *Hypergraph Description*, *Question*, *Tail*, and *Answer*, thereby providing the model with concrete instances to guide its understanding and response generation.

- *COT*: Enhancing the Few-Shot configuration, the CoT setup adds *COT-for-Answer* to each example within the *Example* section. This addition entails detailed explanations of how each answer is systematically derived, enabling the model to follow a logical reasoning pathway for more accurate and transparent responses.

- *COT-HYPER-BAG*: This advanced configuration builds upon the CoT setup by inserting the *Hyper-BAG* component as a transitional sentence between the *Hypergraph Description* and the *Question*. The *Hyper-BAG* serves to contextualize the model's focus on hypergraph construction and comprehension, thereby guiding it into the appropriate analytical framework for handling hypergraph-specific queries.

By integrating these varied prompt structures, the proposed framework effectively leverages existing prompting strategies—such as Chain-of-Thought and Few-Shot Prompting—while introducing hypergraph-specific enhancements. This comprehensive prompt architecture ensures that large language models are adequately guided through diverse hypergraph scenarios, encompassing both synthetic and real-world complexities. Consequently, the framework optimizes the models' interpretative and reasoning capabilities, facilitating a more precise and nuanced evaluation of their hypergraph comprehension within the `LLM4Hypergraph` Benchmark. The deliberate inclusion of optional modules allows for flexible prompt configurations tailored to different evaluation requirements, thereby enhancing the robustness and versatility of the benchmarking process.

## G    Details of Hypergraph Language

Unlike traditional graphs, where an edge connects exactly two vertices, hypergraphs allow hyperedges to connect any number of vertices, thereby necessitating more intricate textual descriptions. However, directly employing abstract language to describe high-order associative structures in hypergraphs can impede LLMs' comprehension of hypergraph architectures, subsequently diminishing performance on downstream tasks. Conversely, utilizing language that describes low-order structures facilitates better understanding by LLMs but may result in the loss of essential high-order information. Inspired by graph description languages, we have designed two distinct types of languages for describing hypergraph structures: *Low-Order Structure Language* and *High-Order Structure Language*, as shown in Figure 4. The former primarily adopts a graph-like approach, leveraging low-order structural descriptions to represent hypergraphs by focusing on pairwise relationships and individual connections. This method enhances LLMs' ability to parse and interpret the hypergraph by simplifying its representation into more familiar graph terminology. In contrast, the High-Order Structure Language directly conveys the complex, multi-vertex correlations inherent in hypergraphs, encapsulating high-order relational information without reducing them to mere pairwise interactions. This approach preserves the richness of hypergraph structures, allowing LLMs to grasp the full extent of multi-way relationships that define hypergraph topology. By employing these two complementary descriptive languages, our framework ensures that LLMs can effectively balance the simplicity of low-order descriptions with the comprehensive detail of high-order structures, thereby optimizing their ability to understand and perform accurately on hypergraph-related tasks within the `LLM4Hypergraph` Benchmark.

### G.1    Low-Order Structure Language

To effectively facilitate LLMs' understanding of hypergraph structures through pairwise correlations, we introduce three distinct low-order structure languages as follows: Low-Order Incidence Description (LO-Inc), Neighbor-Pair Description (N-Pair), and Raw Adjacency Matrix Description (Adj-Mat). Each method provides a unique approach to textualizing hypergraph relationships, enhancing the models' ability to interpret and process hypergraph data.

- *Low-Order-Incidence Description (LO-Inc)*: This method directly describes how a given vertex is connected to other vertices through hyperedges by explicitly stating the connections in a pairwise manner. For example, consider a hypergraph where vertex $v_1$ is connected to vertices $v_2$ and $v_3$. The LO-Inc representation would be: *Vertex $v_1$ is connected to vertices $v_2$ and $v_3$*. This description clearly indicates the direct correlations involving $v_1$, facilitating the model's understanding of its immediate connections within the hypergraph.

- *Neighbor-Pair Description (N-Pair)*: This method focuses on enumerating all pairs of vertices that share a hyperedge, thereby explicitly listing the connected vertex pairs within the hypergraph. Using the same hypergraph example where vertex $v_1$ is connected to vertices $v_2$ and $v_3$, the N-Pair representation would be: $(v_1, v_2), (v_1, v_3)$. This method emphasizes pairwise relationships without referencing the broader connectivity of individual vertices, providing a straightforward enumeration of connected pairs.

- *Raw Adjacency Matrix Description (Adj-Mat)*: This method utilizes an adjacency matrix to represent the hypergraph structure numerically. In this matrix-based approach, the presence or absence of a connection between any two vertices is indicated by binary values. For the hypergraph where vertex $v_1$ is connected to vertices $v_2$ and $v_3$, the Adj-Mat representation would be: $[[0, 1, 1, 0], [1, 0, 1, 1], [1, 1, 0, 0], [0, 1, 0, 0]]$. In this adjacency matrix, each row and column corresponds to a vertex ($v_1$ to $v_4$). A value of '1' at position $(i, j)$ indicates that vertex $v_i$ is connected to vertex $v_j$ via a hyperedge, while a '0' signifies no direct connection. For instance, the entry at $(1, 2)$ is '1', indicating a connection between $v_1$ and $v_2$.

These low-order structure languages—LO-Inc, N-Pair, and Adj-Mat—provide varied methods for representing hypergraph structures through pairwise correlations. *LO-Inc* offers descriptive clarity by outlining direct vertex connections, *N-Pair* emphasizes the enumeration of connected vertex pairs, and *Adj-Mat* delivers a comprehensive numerical representation through an adjacency matrix. By employing these diverse descriptive techniques, our framework enhances LLMs' ability to accurately interpret and analyze the fundamental properties and local relationships within hypergraphs, thereby supporting robust performance on subsequent high-order and isomorphic tasks within the LLM4Hypergraph Benchmark.

## G.2 HIGH-ORDER STRUCTURE LANGUAGE

To further enhance LLMs' comprehension of hypergraph structures through higher-order correlations, we introduce four distinct high-order structure languages as follows: *High-Order Neighbor Description (HO-Neigh)*, *High-Order Incidence Description (HO-Inc)*, *Neighbor-Set Description (N-Set)*, and *Raw Incidence Matrix Description (Inc-Mat)*.

- *High-Order Neighbor Description (HO-Neigh)*: This method leverages the neighbor definitions from HGNN$^+$, describing hypergraph relationships in a two-stage manner by first detailing the connections between vertices and hyperedges, and subsequently outlining the connections between hyperedges and their constituent vertices. For instance, it may state, "Vertex $v_1$ is connected to hyperedge $e_1$", followed by "Hyperedge $e_1$ is connected to vertices $v_1$, $v_2$, and $v_3$", thereby providing a comprehensive view of the local neighborhood structure.

- *High-Order Incidence Description (HO-Inc)*: This method extends the *LO-Inc* by incorporating higher-order correlations between vertices. An example of HO-Inc would be, "Vertex $v_1$ is connected to vertices $v_2$ and $v_3$ with hyperedge $e_1$", which succinctly captures the multi-vertex association facilitated by a single hyperedge.

- *Neighbor-Set Description (N-Set)*: This method builds upon the *N-Pair* by enumerating entire sets of vertices connected by each hyperedge, thus providing a more holistic representation of the hypergraph's connectivity. For example, it may list, "$(v_1, v_2, v_3)$" and "$(v_2, v_4)$", explicitly indicating the groups of vertices that jointly form hyperedges.

- *Raw Incidence Matrix Description (Inc-Mat)*: This method employs an incidence matrix to numerically represent the hypergraph structure, where each entry indicates the presence or absence of a connection between vertices and hyperedges. An example of an incidence matrix is $[[1, 0], [1, 1], [1, 0], [0, 1]]$. In this matrix, rows correspond to vertices ($v_1$ to $v_4$)

and columns correspond to hyperedges ($e_1$ and $e_2$), where a value of 1 denotes the inclusion of a vertex in a hyperedge and 0 denotes absence.

Collectively, these high-order structure languages (HO-Neigh, HO-Inc, N-Set, and Inc-Mat) provide varied and sophisticated methods for representing hypergraph structures through multi-vertex associations. By utilizing these descriptive techniques, our framework facilitates a deeper and more nuanced understanding of hypergraph topologies, thereby supporting advanced analytical capabilities in subsequent high-order and isomorphic tasks within the LLM4Hypergraph Benchmark.

# H  DETAILS OF SPECIFICAL PROMPTING FOR HYPERGRAPHS

Given that hypergraphs do not naturally lend themselves to description using conventional natural language constructs, we introduce two specialized instruction-based prompting techniques designed to enhance LLMs' ability to comprehend hypergraph structures: *Hyper-BAG* and *Hyper-COT*.

- *Build-a-Hypergraph Prompting (Hyper-BAG)* addresses the inherent complexity of hypergraphs by facilitating a mental visualization of their structures (Wang et al., 2024). Recognizing that certain aspects of hypergraph comprehension are more effectively understood through visualization, Hyper-BAG employs instruction-based enhancements that explicitly guide LLMs to imagine the hypergraph's architecture. This approach provides a conceptual buffer, allowing the model to assimilate hypergraph information by mapping it into a structured conceptual space. For instance, an instruction might prompt the model to "imagine a hypergraph where vertex $v_1$ is connected to hyperedge $e_1$," thereby aiding the model in forming a coherent mental representation that better prepares it for subsequent queries related to the hypergraph.

- *Chain-of-Thought for Hypergraphs (Hyper-COT)* builds upon the traditional CoT prompting methodology (Wei et al., 2022) to address the observed difficulties that LLMs encounter in understanding hypergraph structures. Inspired by the step-by-step reasoning inherent in CoT, Hyper-COT integrates a tailored reasoning process specifically designed for hypergraphs. By appending "step-by-step thinking" instructions, Hyper-COT encourages the model to decompose complex hypergraph-related tasks into manageable inference steps, thereby enhancing its comprehension in zero-shot scenarios. Through our experiments, we identified four hypergraph-friendly prompts that effectively guide the model's reasoning process.

  - v1: *"Let's think step by step. Make sure the data is calculated and recorded accurately at each step."*
  - v2: *"Let's analyze the connectivity by considering hyperedges linked to vertices and vertices linked through hyperedges."*
  - v3: *"Let's think hyperedges connected by vertices then vertices connected by hyperedges."*

  These prompts are strategically designed to align the model's reasoning with the multi-way relationships characteristic of hypergraphs, thereby facilitating a deeper and more accurate understanding of hypergraph connectivity and structure.

By implementing Hyper-BAG and Hyper-COT within our prompting framework, we aim to significantly bolster the LLMs' proficiency in interpreting and processing hypergraph structures. These instruction-based techniques address the unique challenges posed by hypergraphs' multi-vertex associations, ensuring that models can effectively navigate both low-order and high-order structural information. Consequently, these strategies play a crucial role in advancing the capabilities of LLMs within the `LLM4Hypergraph` Benchmark, promoting more accurate and sophisticated analyses of hypergraph data.

# I    PROMPT EXAMPLES

## I.1    EXAMPLES OF HYPERGRAPH LANGUAGES

### I.1.1    LOW-ORDER STRUCTURE LANGUAGES

---

**LO-Inc**

**Prompt:** *G describes a hypergraph among vertices v0, v1, v2, v3, v4, and v5 and hyperedges e0, e1, e2, and e3.*
*In this hypergraph:*
*Vertex v0 is connected to vertices v1, v2, v3, v4, v5.*
*Vertex v1 is connected to vertices v0, v2, v3, v4, v5.*
*Vertex v2 is connected to vertices v0, v1, v3, v4, v5.*
*Vertex v3 is connected to vertices v0, v1, v2, v4, v5.*
*Vertex v4 is connected to vertices v0, v1, v2, v3, v5.*
*Vertex v5 is connected to vertices v0, v1, v2, v3, v4.*

---

**N-Pair**

**Prompt:** *In an undirected hypergraph, (i,j) means that vertex i and vertex j are connected with an undirected hyperedge. G describes a hypergraph among vertices v0, v1, v2, v3, v4, and v5 and hyperedges e0, e1, e2, and e3.*
*The connection relation between vertices in G are: (v0, v1) (v2, v4) (v1, v2) (v0, v4) (v3, v4) (v1, v5) (v0, v3) (v1, v4) (v2, v3) (v0, v2) (v4, v5) (v0, v5) (v2, v5) (v1, v3) (v3, v5).*

---

**Adj-Mat**

**Prompt:** *G describes a hypergraph among vertices v0, v1, v2, v3, v4, and v5 and among hyperedges e0, e1, e2, and e3.*
*The adjacency matrix between vertices of the hypergraph is*
*[[1,1,1,1,1,1,], [1,1,1,1,1,1,], [1,1,1,1,1,1,],*
*[1,1,1,1,1,1,], [1,1,1,1,1,1,], [1,1,1,1,1,1,]]*

---

### I.1.2    HIGH-ORDER STRUCTURE LANGUAGES

---

**HO-Neigh**

**Prompt:** *G describes a hypergraph among vertices v0, v1, v2, v3, v4, and v5 and hyperedges e0, e1, e2, and e3.*
*In this hypergraph:*
*Vertex v0 is connected to hyperedges e1, e2.*
*Vertex v1 is connected to hyperedges e0, e1,e3.*
*Vertex v2 is connected to hyperedges e0, e1.*
*Vertex v3 is connected to hyperedges e0, e1.*
*Vertex v4 is connected to hyperedges e0, e1, e2.*
*Vertex v5 is connected to hyperedges e1, e3.*
*Hyperedge e0 is connected to vertices v1, v2, v3, v4.*
*Hyperedge e1 is connected to vertices v0, v1, v2, v3, v4, v5.*
*Hyperedge e2 is connected to vertices v0, v4.*
*Hyperedge e3 is connected to vertices v1, v5.*

---

### HO-Inc

**Prompt:** *G describes a hypergraph among vertices v0, v1, v2, v3, v4, and v5 and among hyperedges e0, e1, e2, and e3.*
*In this hypergraph:*
*Vertex v0 is connected to vertices v1, v2, v3, v4, v5 with hyperedge e1, to vertex v4 with hyperedge e2.*
*Vertex v1 is connected to vertices v2, v3, v4 with hyperedge e0, to vertices v0, v2, v3, v4, v5 with hyperedge e1, to vertex v5 with hyperedge e3.*
*Vertex v2 is connected to vertices v1, v3, v4 with hyperedge e0, to vertices v0, v1, v3, v4, v5 with hyperedge e1.*
*Vertex v3 is connected to vertices v1, v2, v4 with hyperedge e0, to vertices v0, v1, v2, v4, v5 with hyperedge e1.*
*Vertex v4 is connected to vertices v1, v2, v3 with hyperedge e0, to vertices v0, v1, v2, v3, v5 with hyperedge e1, to vertex v0 with hyperedge e2.*
*Vertex v5 is connected to vertices v0, v1, v2, v3, v4 with hyperedge e1, to vertex v1 with hyperedge e3.*

### N-Set

**Prompt:** *In an undirected hypergraph, (i, j, k) means that vertex i, vertex j and vertex k are connected with an undirected hyperedge. G describes a hypergraph among vertices v0, v1, v2, v3, v4, and v5, and among hyperedges e0, e1, e2, and e3.*
*The hyperedges in G are: (v1, v2, v3, v4), (v0, v1, v2, v3, v4, v5), (v0, v4), (v1, v5).*

### Inc-Mat

**Prompt:** *G describes a hypergraph among vertices v0, v1, v2, v3, v4, and v5 and hyperedges e0, e1, e2, and e3.*
*The incidence matrix of the hypergraph is*
*[[0,1,1,0,],*
*[1,1,0,1,],*
*[1,1,0,0,],*
*[1,1,0,0,],*
*[1,1,1,0,],*
*[0,1,0,1,]]*

## I.2 EXAMPLES OF QUESTION FOR DIFFERENT TASKS

### I.2.1 LOW-ORDER TASKS

### Hyperedge Count

**Prompt:** *How many hyperedges are in this hypergraph?*
*List the answers after "Ans:" in the format like [10].*

### Vertex Count

**Prompt:** *How many vertices are in this hypergraph?*
*List the answers after "Ans:" in the format like [10].*

### Vertex Degree

**Prompt:** *What is the degree of vertex v14?*
*List the answers after "Ans:" in the format like [10].*

### Vertex Connection Check

**Prompt:** *Is vertex v14 connected to vertex v3?*
*List the answers after "Ans:" in the format of [Yes, No,].*

### Reachability Check

**Prompt:** *Is there a path from node v0 to node v1?*
*List the answers after "Ans:" in the format of [Yes, No,].*

### Shortest Path

**Prompt:** *What is the length of the shortest path from node v0 to node v5?*
*List the answers after "Ans:" in the format of [Yes, No,].*

### Connected Vertices

**Prompt:** *List all the vertices connected to v14 in alphabetical order.*
*List all the answers after "Ans:" in the format of [v0, v1, v2] and separate the answers by a comma.*

### Disconnected Vertices

**Prompt:** *List all the vertices that are not connected to v14 in alphabetical order.*
*List all the answers after "Ans:" in the format of [v0, v1, v2] and separate the answers by a comma.*

### I.2.2 HIGH-ORDER TASKS

### Hyperedge Degree

**Prompt:** *What is the degree of hyperedge e4?*
*List the answers after "Ans:" in the format like [10].*

### Vertex Set Connection Check

**Prompt:** *Is there a hyperedge that contain both vertex set (v2, v3) and vertex set (v5, v7)?*
*List the answers after "Ans:" in the format of [Yes, No,].*

### Vertex-Set-in-Hyperedge Check

**Prompt:** *Is there a hyperedge that contains all vertices in vertex set (v4, v16)?*
*List the answers after "Ans:" in the format of [Yes, No,].*

> **Hyperedge-in-Hyperedge Check**
>
> **Prompt:** *Whether any hyperedge in the hypergraph is contained by other hyperedges?*
> *List the answers after "Ans:" in the format of [Yes, No,].*

> **Shared-Vertices between Hyperedges**
>
> **Prompt:** *List the vertices connected to both hyperedge e3 and hyperedge e2 in alphabetical order.*
> *List all the answers after "Ans:" in the format of [v0, v1, v2] and separate the answers by a comma.*

### I.2.3  ISOMORPHISM TASKS

> **Isomorphism Recognition**
>
> **Prompt:** *Are these two hypergraphs isomorphism?*
> *List the answers after "Ans:" in the format of [Yes, No,].*

> **Structure Classification**
>
> **Prompt:** *What is the shape of the visualization of the hypergraph like? Please directly give the answer number corresponding to the 3 hypergraph visualization shapes as shown below.*
> *Answer [0] corresponds to the Hyper Pyramid, which represents the visualization of the hypergraph as a Pyramid.*
> *Answer [1] corresponds to the Hyper Checked Table.*
> *Answer [2] corresponds to the Hyper Wheel.*
> *List the answer after "Ans:" in the format like [2].*

### I.3  EXAMPLES OF PROMPTS UNDER DIFFERENT SETTINGS

> **ZERO SHOT**
>
> **Prompt:** *In an undirected hypergraph, (i, j, k) means that vertex i, vertex j, and vertex k are connected with an undirected hyperedge. G describes a hypergraph among vertices v0, v1, v2, v3, v4, and v5 and among hyperedges e0, e1, e2, and e3.*
> *The hyperedges in G are: (v1, v2, v3, v4), (v0, v1, v2, v3, v4, v5), (v0, v4), (v1, v5).*
> *Let's construct the hypergraph with the vertices and hyperedges first.*
> *Q: How many hyperedges are in this hypergraph? list the answers after "Ans:" in the format like [10].*
> *A:*
> - - - - - - - - - - - - - - - - - - - - - - - - - - - - - - - - - - - - - - - -
> **Response:** *Ans:[4]*
>
> **Ground Truth:** *Ans:[4]*

---

**ZERO-HYPER-COT**

**Prompt:** *In an undirected hypergraph, (i, j, k) means that vertex i, vertex j and vertex k are connected with an undirected hyperedge. G describes a hypergraph among vertices v0, v1, v2, v3, v4, and v5 and among hyperedges e0, e1, e2, and e3.*
*The hyperedges in G are: (v1, v2, v3, v4), (v0, v1, v2, v3, v4, v5), (v0, v4), (v1, v5).*
*Let's construct the hypergraph with the vertices and hyperedges first.*
*Q: How many hyperedges are in this hypergraph? list the answers after "Ans:" in the format like [10].*
*A: Let's think hyperedges connected by vertices then vertices connected by hyperedges.*

- - - - - - - - - - - - - - - - - - - - - - - - - - - - - - - - - - - - - - - - - - - - - -

**Response:** *To determine the number of hyperedges in the given hypergraph G, we need to count each unique hyperedge listed.*
*Given hyperedges:*
*1. (v1, v2, v3, v4)*
*2. (v0, v1, v2, v3, v4, v5)*
*3. (v0, v4)*
*4. (v1, v5)*
*Let's count these:*
*1. (v1, v2, v3, v4) - This is one hyperedge connecting vertices v1, v2, v3, and v4 .*
*2. (v0, v1, v2, v3, v4, v5) - This is another hyperedge connecting vertices v0, v1, v2, v3, v4, and v5 .*
*3. (v0, v4) - This is a third hyperedge connecting vertices v0 and v4 .*
*4. (v1, v5) - This is a fourth hyperedge connecting vertices v1 and v5 .*
*So, there are four distinct hyperedges listed.*
*Ans: [4]*

**Ground Truth:** *Ans:[4]*

---

**FEW SHOT**

**Prompt:** *Example: In an undirected hypergraph, (i, j, k) means that vertex i, vertex j and vertex k are connected with an undirected hyperedge. G describes a hypergraph among vertices v0, v1, v2, v3, v4, v5, v6, v7, v8, v9, v10, v11, v12, v13, v14, v15, v16, and v17 and among hyperedges e0, e1, e2, e3, e4, e5, e6, e7, e8, e9, e10, e11, e12, e13, e14, e15, e16, e17, e18, and e19.*
*The hyperedges in G are: (v8, v16, v17), (v11, v14, v17), (v4, v15), (v1, v6, v15, v17), (v0, v11), (v2, v14), (v6, v11), (v3, v9, v15), (v5, v9),(v9, v10), (v7, v8, v15), (v0, v10), (v1, v11), (v7, v9), (v15, v16), (v9, v11, v12), (v4, v13), (v9, v12), (v2, v12), (v7, v14).*
*Q: How many hyperedges are in this hypergraph? list the answers after "Ans:" in the format like [10].*
*A: Ans:[20].*

*In an undirected hypergraph, (i, j, k) means that vertex i, vertex j and vertex k are connected with an undirected hyperedge. G describes a hypergraph among vertices v0, v1, v2, v3, v4, and v5 and among hyperedges e0, e1, e2, and e3.*
*The hyperedges in G are: (v1, v2, v3, v4), (v0, v1, v2, v3, v4, v5), (v0, v4), (v1, v5).*
*Q: How many hyperedges are in this hypergraph? list the answers after "Ans" in the format like [10].*
*A:*

- - - - - - - - - - - - - - - - - - - - - - - - - - - - - - - - - - - - - - - - - - - - - -

**Response:** *Ans:[4]*

**Ground Truth:** *Ans:[4]*

## COT

**Prompt:** *Example: In an undirected hypergraph, (i, j, k) means that vertex i, vertex j and vertex k are connected with an undirected hyperedge. G describes a hypergraph among vertices v0, v1, v2, v3, v4, v5, v6, v7, v8, v9, v10, v11, v12, v13, v14, v15, v16, and v17 and among hyperedges e0, e1, e2, e3, e4, e5, e6, e7, e8, e9, e10, e11, e12, e13, e14, e15, e16, e17, e18, and e19.*
*The hyperedges in G are: (v8, v16, v17), (v11, v14, v17), (v4, v15), (v1, v6, v15, v17), (v0, v11), (v2, v14), (v6, v11), (v3, v9, v15), (v5, v9), (v9, v10), (v7, v8, v15), (v0, v10), (v1, v11), (v7, v9), (v15, v16), (v9, v11, v12), (v4, v13), (v9, v12), (v2, v12), (v7, v14).*
*Q: How many hyperedges are in this hypergraph? list the answers after "Ans:" in the format like [10].*
*A: Ans:[20]. The hyperedges are e0, e1, e2, e3, e4, e5, e6, e7, e8, e9, e10, e11, e12, e13, e14, e15, e16, e17, e18, and e19.*

*In an undirected hypergraph, (i, j, k) means that vertex i, vertex j and vertex k are connected with an undirected hyperedge. G describes a hypergraph among vertices v0, v1, v2, v3, v4, and v5 and among hyperedges e0, e1, e2, and e3.*
*The hyperedges in G are: (v1, v2, v3, v4), (v0, v1, v2, v3, v4, v5), (v0, v4), (v1, v5).*
*Q: How many hyperedges are in this hypergraph? list the answers after "Ans" in the format like [10].*
*A:*

- - - - - - - - - - - - - - - - - - - - - - - - - - - - - - - - - - - - - - - - -

**Response:** *Ans:[4]. The hyperedges are e0, e1, e2, e3.*

**Ground Truth:** *Ans:[4]*

## COT-HYPER-BAG

**Prompt:** *Example: In an undirected hypergraph, (i, j, k) means that vertex i, vertex j and vertex k are connected with an undirected hyperedge. G describes a hypergraph among vertices v0, v1, v2, v3, v4, v5, v6, v7, v8, v9, v10, v11, v12, v13, v14, v15, v16, and v17 and among hyperedges e0, e1, e2, e3, e4, e5, e6, e7, e8, e9, e10, e11, e12, e13, e14, e15, e16, e17, e18, and e19.*
*The hyperedges in G are: (v8, v16, v17), (v11, v14, v17), (v4, v15), (v1, v6, v15, v17), (v0, v11), (v2, v14), (v6, v11), (v3, v9, v15), (v5, v9), (v9, v10), (v7, v8, v15), (v0, v10), (v1, v11), (v7, v9), (v15, v16), (v9, v11, v12), (v4, v13), (v9, v12), (v2, v12), (v7, v14).*
*Let's construct the hypergraph with the vertices and hyperedges first.*
*Q: How many hyperedges are in this hypergraph? list the answers after "Ans:" in the format like [10].*
*A: Ans:[20]. The hyperedges are e0, e1, e2, e3, e4, e5, e6, e7, e8, e9, e10, e11, e12, e13, e14, e15, e16, e17, e18, and e19.*

*In an undirected hypergraph, (i, j, k) means that vertex i, vertex j and vertex k are connected with an undirected hyperedge. G describes a hypergraph among vertices v0, v1, v2, v3, v4, and v5 and among hyperedges e0, e1, e2, and e3.*
*The hyperedges in G are: (v1, v2, v3, v4), (v0, v1, v2, v3, v4, v5), (v0, v4), (v1, v5).*
*Let's construct the hypergraph with the vertices and hyperedges first.*
*Q: How many hyperedges are in this hypergraph? list the answers after "Ans" in the format like [10].*
*A:*

- - - - - - - - - - - - - - - - - - - - - - - - - - - - - - - - - - - - - - - - -

**Response:** *Ans:[4]. The hyperedges are e0, e1, e2, e3.*

**Ground Truth:** *Ans:[4]*

