# OpenReview forum: "Beyond Graphs: Can Large Language Models Comprehend Hypergraphs?"
_ICLR.cc/2025/Conference — ICLR 2025 Poster_

### Official Review · Reviewer_rA8g · 2024-10-16

**Soundness:** 3
**Presentation:** 3
**Contribution:** 3
**Rating:** 8
**Confidence:** 4

**Summary:**

This paper introduces LLM4Hypergraph, the first benchmark aimed at evaluating the ability of LLMs to understand hypergraph data. The authors design a series of tasks of varying difficulty levels and evaluate six different LLMs. Then, they identify their strengths and weaknesses. While this work represents a first step and provides a comprehensive study, there are several areas where improvement is needed.

**Strengths:**

- This paper proposes the first benchmark for evaluating LLMs on hypergraphs.
- The authors thoroughly address questions about hypergraphs.
- The problems are well-structured and clearly categorized according to their objectives.
- The code is released for reproducibility.

**Weaknesses:**

- The motivations for this research are not sufficiently discussed. Why is it important to enable LLMs to understand hypergraph structures? Are there potential practical use cases? Are there any motivations beyond the fact that similar research has been done with graphs?
- The datasets used in the study are not comprehensive. To be specific:
  - The definition of "hypergraph size" is unclear. Is it referring to the number of nodes, the number of hyperedges, or the sum of hyperedge sizes?
  - The specific sizes of the hypergraphs (both real-world and synthetic) are not mentioned in the main content. How large are the synthetic hypergraphs used for evaluation?
  - According to the appendix, even the so-called "large-scale hypergraphs" only contain 15 to 20 vertices, which is too small to meaningfully capture higher-order structures typically expected in hypergraphs.
  - The synthetic hypergraphs are not sufficiently representative. There are other synthetic hypergraph models (e.g., configuration models) available.
  - It is unclear how the random walk approach for sampling sub-hypergraphs from real-world hypergraphs (Appendix A.2) ensures that the sampled hypergraphs "retain the intricate and authentic correlations inherent in the original data."
- The definition of task "difficulty" is unclear.
- The authors may consider discussing/citing the recent work "When LLM Meets Hypergraph: A Sociological Analysis on Personality via Online Social Networks" (CIKM 2024) in the related work.

**In summary**, this paper makes a valuable contribution to LLMs and hypergraph analysis. However, the benchmark datasets lack comprehensiveness and have room to consider additional synthetic hypergraph generators. Also, the paper lacks detailed statistics on real-world hypergraphs. Scalability is also a concern; if large-scale hypergraph handling poses challenges for LLMs, these limitations should be clearly discussed.

**Questions:**

- How does the performance of LLMs depend on the hypergraph domains (e.g., emails, coauthorship)?

---

> ### Author Response · Authors · 2024-11-15
> **Response to Reviewer rA8g (Part 1/6)**
>
> ### Response to W1
>
> Thanks for your reviews. We would like to further clarify and elaborate on our research motivations.
>
> 1. Motivation for Researching Hypergraphs
>
> Graph representation and learning[1] have been pivotal research directions in computer science. Notably, this year’s Nobel Prize in Chemistry was awarded to AlphaFold[2], whose achievements heavily relied on graph-based representations and computations of protein three-dimensional structures, demonstrating the powerful capabilities of graphs in modeling complex data. However, traditional graph models primarily focus on pairwise relationships, making it challenging to capture more intricate high-order correlations present in real-world data.
>
> Hypergraphs, as a mathematical generalization of graphs, allow a hyperedge to connect more than two nodes, thereby enabling the representation of more complex data associations. High-order correlations in hypergraphs hold significant real-world importance across various domains:
>
> - **Social Networks**: A community or group inherently represents a high-order relationship[3] with an uncertain number of connected nodes, which cannot be effectively modeled using only pairwise edges. In contrast, a hyperedge in a hypergraph can naturally connect any number of nodes, accurately reflecting community relationships.
>
> - **Life Sciences**: In the field of protein structures, catalytic triplets are typical high-order structures involving interactions among three nodes, present in important protein catalysts such as trypsin[4], playing a crucial role in protein degradation processes. Such high-order structures are essential for understanding biological processes, yet traditional graph models struggle to effectively represent and handle these complex relationships.
>
> These real-world hypergraph structures motivate us to delve deeper into hypergraph research to better understand and apply high-order associations.
>
> 2. Motivation for Using LLMs for Hypergraph Understanding
>
> While hypergraph neural networks (HGNNs)[5] can address tasks within hypergraphs, such as node classification and link prediction, these methods often require designing customized model architectures for different tasks, increasing the complexity of model design and limiting their applicability across multiple tasks.
>
> LLMs have recently demonstrated powerful knowledge bases and reasoning capabilities, showing potential for handling diverse tasks. By textualizing hypergraphs, we can leverage the general capabilities of LLMs to handle various hypergraph tasks without the need for designing specialized model structures for each task. This approach offers several advantages:
>
> - **Uniformity and Generality**: A single method (textualization) can be applied to multiple hypergraph tasks, simplifying the model design process and enhancing the method’s generality and adaptability.
>
> - **Cross-Domain Knowledge Integration**: LLMs possess extensive knowledge bases, enabling the integration and transfer of knowledge across different domains, thereby enhancing hypergraph task comprehension and reasoning.
>
> - **Efficient Multi-Task Processing**: By designing different prompts, LLMs can be flexibly guided to accomplish various hypergraph tasks, such as structure recognition, link prediction, and isomorphism detection.
>
> Therefore, our research is not only inspired by similar graph studies but also driven by the critical importance of hypergraphs in various real-world applications, exploring how to harness the powerful capabilities of LLMs to enhance hypergraph understanding and application potential. We will further expand and clarify these motivations in the revised manuscript to strengthen the discussion and persuasiveness of our paper.
>
>
> [1] Kipf T N, Welling M. Semi-supervised classification with graph convolutional networks[J]. arXiv preprint arXiv:1609.02907, 2016.
>
> [2] Jumper J, Evans R, Pritzel A, et al. Highly accurate protein structure prediction with AlphaFold[J]. **Nature**, 2021, 596(7873): 583-589.
>
> [3] Contisciani M, Battiston F, De Bacco C. Inference of hyperedges and overlapping communities in hypergraphs[J]. **Nature Communications**, 2022, 13(1): 7229.
>
> [4] Ravetz B D, Pun A B, Churchill E M, et al. Photoredox catalysis using infrared light via triplet fusion upconversion[J]. **Nature**, 2019, 565(7739): 343-346.
>
> [5] Feng Y, You H, Zhang Z, et al. Hypergraph neural networks[C]//Proceedings of the AAAI conference on artificial intelligence. 2019, 33(01): 3558-3565. **1400+ Citation**

---

> ### Author Response · Authors · 2024-11-15
> **Response to Reviewer rA8g (Part 2/6)**
>
> ### Response to W2.1 and W2.2
>
> Thanks for your reviews. Regarding your concerns about the unclear definition of "hypergraph size" and the lack of specific sizes for both real-world and synthetic hypergraphs used in our experiments, we provide the following detailed clarification.
>
> In our experiments, we define the **size of a hypergraph as the number of nodes** it contains, following the conventions outlined in related literature[1]. Based on the number of nodes, we categorize hypergraphs into three size classes:
>
> 1. **Small-scale Hypergraphs**: Containing 5-10 nodes
> 2. **Medium-scale Hypergraphs**: Containing 10-15 nodes
> 3. **Large-scale Hypergraphs**: Containing 15-20 nodes
>
> Synthetic Hypergraphs
>
> For synthetic hypergraphs, we utilize the `dhg.random.hypergraph_Gnm()` function from the DHG library to generate hypergraphs with a specified number of nodes. This function allows us to create hypergraphs of varying scales to meet different experimental requirements.
>
> Real-world Hypergraphs
>
> Our real-world datasets are derived from two categories:
> 1. **Citation Network Dataset**: We randomly sample sub-hypergraph structures from the CoauthorshipCora dataset[2]. In these hypergraphs, **nodes** represent papers, and **hyperedges** signify co-authorship relationships.
> 2. **Protein Structure Dataset**: Sourced from the Protein Data Bank (PDB)[3], we randomly select proteins and sample their substructures. In these hypergraphs, **nodes** represent protein residues, and **hyperedges** connect residues that have their **alpha carbon atoms within 10 Å** of each other, centered around each residue.
>
> To generate subgraphs of varying sizes (with node counts ranging from 5 to 20), we employ a **random walk** sampling method as follows:
> - During each random walk, each traversed node has a **0.5 probability** of being retained or discarded.
> - Once the predefined number of nodes is reached, all hyperedges associated with these nodes are retained to form the final subgraph hyperedges.
>
>
> We will incorporate the above detailed descriptions of hypergraph size definitions, the specific scales of synthetic and real-world hypergraphs, and the sampling methods used into the revised version of our manuscript. This will ensure that readers have a clear understanding of the sources and construction methods of the datasets used in our experiments.
>
>
> [1] Fatemi B, Halcrow J, Perozzi B. Talk like a Graph: Encoding Graphs for Large Language Models[C]//NeurIPS 2023 Workshop: New Frontiers in Graph Learning.
>
> [2] Yadati N, Nimishakavi M, Yadav P, et al. Hypergcn: A new method for training graph convolutional networks on hypergraphs[J]. Advances in neural information processing systems, 2019, 32.
>
> [3] Bank P D. Protein data bank[J]. Nature New Biol, 1971, 233(223): 10-1038.

---

> ### Author Response · Authors · 2024-11-15
> **Response to Reviewer rA8g (Part 3/6)**
>
> ### Response to W2.3
>
> Thanks for your comments. Regarding your concern that “the so-called ‘large-scale hypergraphs’ in the appendix contain only 15 to 20 vertices, which is too small to meaningfully capture the higher-order structures typically expected in hypergraphs,” we provide the following detailed response.
>
> 1. **Context Length Limitations of Prompting**: Current large language models (LLMs) have limitations on the context length they can process. Although hypergraphs themselves can represent extremely complex high-order relationships, the context window constraints necessitated our choice of hypergraphs with 15 to 20 nodes to ensure that all structural information of the hypergraphs could be fully inputted and processed. This selection was made to comprehensively evaluate the LLM's ability to understand and reason about hypergraph structures within the model’s current capabilities.
>
> 2. **Complexity of Hypergraphs Relative to Node Count**: In the context of hypergraphs, increasing the number of nodes exponentially increases the potential number of hyperedges. Specifically, a hypergraph with 20 nodes can have up to 2^20 possible hyperedges, which far surpasses the 20^2 edges typical in traditional graphs. This implies that even with a relatively small number of nodes, hypergraphs can exhibit extremely high complexity and diversity in their high-order structures. Therefore, despite the seemingly limited node count in our "large-scale hypergraphs," the complexity and richness of the high-order relationships provide a substantial basis for effectively assessing the LLM's understanding and reasoning capabilities.
>
> 3. **Complexity of Hypergraph Tasks and Future Research Directions**: Hypergraph tasks inherently possess greater complexity and challenge, especially when dealing with larger-scale hypergraphs. Currently, we have chosen a node range of 15-20 to balance complexity and manageability, ensuring the feasibility and validity of our experiments. However, as technology advances and model capabilities improve, we plan to explore and handle larger-scale hypergraphs in future work to further validate and expand the application potential of LLMs in understanding high-order structures.
>
> In summary, although our current "large-scale hypergraphs" have a limited number of nodes, their high-order structural complexity provides sufficient challenge for evaluating the capabilities of LLMs. We will further clarify this point in the revised manuscript and aim to expand the scale of hypergraphs in our future research.
>
>
> ### Response to W2.4
>
>
> Thanks for your comments. Regarding your concern that “the synthetic hypergraphs are not sufficiently representative and that other synthetic hypergraph models (e.g., configuration models) are available,” we provide the following detailed response.
>
> In this study, we employ a **Low-Order First** approach to generate synthetic hypergraphs. This method assumes that real-world associations prioritize simpler relationships, aligning with the principle of Occam’s Razor, which suggests that among competing hypotheses, the one with the fewest assumptions should be selected. Therefore, the Low-Order First approach effectively simulates and represents the common simple association structures found in real-world scenarios.
>
> To encompass different types of typical hypergraph structures, we utilize three representative structures in our synthetic hypergraph generation process:
>
> 1. **Hyper Pyramid**: Simulates hierarchical high-order relationships, akin to multi-layered connections in a pyramid structure.
> 2. **Hyper Checked Table**: Simulates grid-like high-order relationships, similar to the intersecting connections in a table grid.
> 3. **Hyper Wheel**: Simulates wheel-like high-order relationships, resembling spokes connecting to a central hub node.
>
> These structures are prevalent in various application contexts, representing diverse high-order association patterns and ensuring diversity and representativeness in the structural composition of the synthetic hypergraphs.
>
>
> While the current synthetic hypergraph models effectively simulate various typical structures, we acknowledge the importance of incorporating additional synthetic hypergraph models (such as configuration models) to capture more complex and diverse high-order associations. Therefore, in future research, we plan to introduce a wider variety of synthetic hypergraph models, including configuration models, to further enhance the representativeness and comprehensiveness of our benchmark. This will allow for a more thorough evaluation of large language models’ capabilities in understanding and reasoning across different types of hypergraphs.
>
>
> We will outline our plans to incorporate more synthetic hypergraph models in future work in response to your suggestion. This will address your concerns and strengthen the completeness and persuasiveness of our benchmark.

---

> ### Author Response · Authors · 2024-11-15
> **Response to Reviewer rA8g (Part 4/6)**
>
> ### Response to W2.5
>
> Thanks for your comments. Regarding your concern that “it is unclear how the random walk approach for sampling sub-hypergraphs from real-world hypergraphs (Appendix A.2) ensures that the sampled hypergraphs 'retain the intricate and authentic correlations inherent in the original data,'” we would like to provide a detailed response.
>
> In our experiments, to sample sub-hypergraphs of varying sizes (ranging from 5 to 20 nodes) from real-world hypergraphs, we employed a random walk-based approach. The specific steps are as follows:
>
> 1. **Random Walk Process**:
>     - We initiate the random walk from a randomly selected node within the hypergraph.
>     - At each step of the walk, each neighboring node of the current node has a 50% probability of being retained or discarded.
>     - This process continues until the predefined number of nodes (between 5 and 20) is reached.
>
> 2. **Retention of Hyperedges**:
>     - Once the node set for the sub-hypergraph is determined, we retain all hyperedges that are associated with these selected nodes.
>     - These hyperedges capture the high-order relationships between the nodes, ensuring that the complex structural information of the original hypergraph is preserved within the sub-hypergraph.
>
> 3. **Ensuring Authentic High-Order Structures**:
>     - By sampling around specific nodes and their respective domains, the random walk method effectively captures the actual association patterns present in the original hypergraph.
>     - This approach not only preserves direct relationships between nodes but also maintains the high-order associations represented by hyperedges, ensuring that the sub-hypergraphs retain the intricate and authentic correlations inherent in the original data.
>
> Advantages of the Sampling Method
>
> - **Structural Diversity**: The random walk method allows for coverage of different parts of the hypergraph, resulting in sub-hypergraphs with diverse topological structures.
> - **Association Retention**: By retaining all relevant hyperedges, the method ensures that the high-order relationships within the sub-hypergraphs reflect those in the original hypergraphs, maintaining data complexity and authenticity.
> - **Scalability**: This approach is versatile and applicable to various sizes and types of hypergraphs, offering good generality and flexibility.
>
>
> To address your suggestion, we will include a detailed description of the random walk sampling method and how it effectively preserves the high-order association structures from the original hypergraphs in the revised manuscript. Through these additions, we aim to clearly illustrate how our sampling method ensures that the sub-hypergraphs retain the intricate and authentic correlations inherent in the original data.
>
>
> ### Response to W4
>
> Thanks for your comments. You recommended that we discuss and cite the recent work "When LLM Meets Hypergraph: A Sociological Analysis on Personality via Online Social Networks" (CIKM 2024) in the related work section. We acknowledge that this paper is a valuable addition to our research.
>
> This paper leverages Large Language Models (LLMs) to enhance node representations by integrating various dispersed information and external knowledge, thereby generating higher-quality data and significantly improving the performance of personality analysis. Additionally, the authors utilize Hypergraph Neural Networks (HGNNs) to analyze social networks, enabling the identification of human personalities. This aligns with our work in utilizing LLMs to understand hypergraph structures, both demonstrating the potential and advantages of LLMs in handling complex high-order relational data.
>
> To further enrich our related work section, we will include a discussion and citation of this paper in the revised manuscript.

---

> ### Author Response · Authors · 2024-11-15
> **Response to Reviewer rA8g (Part 5/6)**
>
> ### Response to W5 (Summary)
>
> Thanks for your comments. We have the following response to your summary.
>
> #### Regarding "Additional Synthetic Hypergraph Generators"
>
> In the current version, we have constructed a preliminary benchmark that includes over 20,000 hypergraph question-answer (Q-A) samples and 15 hypergraph tasks. To ensure diversity and representativeness within this benchmark, we employ four hypergraph generation methods:
> 1. **Low-Order First Random Hypergraphs**: This approach assumes that real-world associations prioritize simple connections, aligning with the principle of Occam’s razor.
> 2. **Three Types of Structured Hypergraphs**:
>    - **Hyper Pyramid**: Simulates hierarchical high-order relationships.
>    - **Hyper Checked Table**: Simulates grid-like high-order relationships.
>    - **Hyper Wheel**: Simulates wheel-like high-order relationships.
>
> These generation methods cover a range of typical high-order association patterns, ensuring structural diversity within the synthetic hypergraphs. However, we acknowledge that our current synthetic hypergraph models are not exhaustive. In future work, we plan to incorporate additional hypergraph generation models to further enhance the comprehensiveness and representativeness of our benchmark. This expansion will allow for a more robust evaluation of LLMs across diverse hypergraph structures.
>
> #### Regarding "Detailed Statistics on Real-World Hypergraphs"
>
> To enhance the transparency and comprehensibility of our study, we have conducted detailed statistical analyses of the real-world hypergraphs used in our experiments. The results are summarized below:
>
> |              | Averaged Number of Vertices | Averaged Number of Hyperedge | Averaged Vertex Degree | Averaged Hyperedge Degree |
> |--------------|:---------------------------:|:----------------------------:|:----------------------:|:-------------------------:|
> | Coauthorship |            11.75            |             4.58             |          1.30          |            3.34           |
> |    Protein   |            10.06            |             16.55            |          4.53          |            2.75           |
>
> We will include these detailed statistics in the revised manuscript.
>
> #### Regarding "Scalability and Large-Scale Hypergraph Handling Challenges"
>
> 1. **Context Length Limitations of Prompting**: Current large language models (LLMs) have inherent limitations regarding the context length they can process. While hypergraphs can represent highly complex high-order relationships, the constraints of the context window necessitated our selection of hypergraphs with node counts ranging from 15 to 20. This ensures that all structural information of the hypergraphs can be fully inputted and processed by the models. This selection was made to balance the model’s capacity with the need to comprehensively evaluate the LLM’s understanding and reasoning capabilities within these constraints.
>
> 2. **Complexity of Hypergraphs Relative to Node Count**: In hypergraphs, increasing the number of nodes exponentially increases the potential number of hyperedges. Specifically, a hypergraph with 20 nodes can have up to $2^{20}$ possible hyperedges, which far surpasses the $20^2$ edges typical in traditional graphs. This means that even with a relatively small number of nodes, hypergraphs can exhibit extreme complexity and diversity in their high-order structures. Therefore, despite the seemingly limited node count in our "large-scale hypergraphs," the complexity and richness of the high-order relationships provide a substantial basis for effectively assessing the LLM’s understanding and reasoning capabilities.
>
> 3. **Complexity of Hypergraph Tasks and Future Research Directions**: Hypergraph tasks inherently possess greater complexity and challenge, especially when dealing with larger-scale hypergraphs. Currently, we have chosen a node range of 15-20 to balance complexity and manageability, ensuring the feasibility and validity of our experiments. However, as technology advances and model capabilities improve, we plan to explore and handle larger-scale hypergraphs in future work to further validate and extend the application potential of LLMs in understanding high-order structures.

---

> ### Author Response · Authors · 2024-11-15
> **Response to Reviewer rA8g (Part 6/6)**
>
> ### Response to Q1
>
> Thanks for your suggestions. In response to your query on how the performance of Large Language Models (LLMs) depends on different hypergraph domains, we further conduct an additional set of experiments to evaluate the performance of LLMs across various hypergraph datasets.
>
> In our supplementary experiments, we selected representative tasks to assess both low-order and high-order understanding capabilities of LLMs. Specifically, we chose the **Vertex Connection Check Task** as a typical low-order node understanding task and the **Vertex-Set-in-Hyperedge Check Task** as a representative high-order hyperedge understanding task. To encode the hypergraph structures, we employ four distinct hypergraph high-order encoding methods, ensuring a comprehensive evaluation of different encoding strategies. These experiments were conducted on two real-world datasets: the **Coauthorship dataset** and the **Protein dataset**. Upon statistical analysis, we find that the Coauthorship dataset has an average hyperedge degree of 3.34, while the Protein dataset has an average hyperedge degree of 2.75. For comparison, classic low-order structured graphs typically exhibit an average hyperedge degree of 2. This metric allow us to quantify the high-order nature of each dataset, with the Coauthorship dataset demonstrating a significantly higher order.
>
>
>
> |                           |      Low-Order Task     |      Low-Order Task     |        High-Order Task        |        High-Order Task        |
> |:-------------------------:|:-----------------------:|:-----------------------:|:-----------------------------:|:-----------------------------:|
> |                           | Vertex Connection Check | Vertex Connection Check | Vertex-Set-in-Hyperedge Check | Vertex-Set-in-Hyperedge Check |
> |                           |       Coauthorship      |         Protein         |          Coauthorship         |            Protein            |
> | Averaged Hyperedge Degree |           3.34          |           2.75          |              3.34             |              2.75             |
> |           N-Set           |           0.98          |          0.976          |             0.956             |             0.946             |
> |           HO-Inc          |          0.996          |          0.996          |             0.972             |             0.872             |
> |           Inc-Mat         |          0.798          |          0.728          |             0.912             |             0.946             |
> |          HO-Neigh         |          0.764          |          0.876          |             0.884             |             0.816             |
> |      Averaged Results     |          0.884         |        **0.894**        |           **0.931**           |             0.895             |
>
>
> Our experimental results reveal that for the high-order **Vertex-Set-in-Hyperedge Check Task**, the performance of LLMs improved as the hyperedge degree of the dataset increased, with an approximate enhancement of 3.6%. This indicates that our proposed high-order encoding methods effectively boost the LLMs' ability to handle complex high-order relational tasks, particularly benefiting from the richer high-order structures present in the Coauthorship dataset.
> For the low-order **Vertex Connection Check Task**, we observe that the lower the hyperedge degree of the dataset, the better the LLM's performance. This is because lower-order datasets have structures that are closer to traditional graph structures, making high-order descriptive language more redundant. Consequently, this redundancy makes it more difficult for the LLM to comprehend the structure, leading to a decline in performance.
>
>
> Furthermore, our findings confirm that variations in domains indeed lead to differences in LLM performance. These differences are fundamentally due to variations in hypergraph data distributions across domains, such as the sparsity of connections and the number of hyperedges. To gain a deeper and more comprehensive understanding of these domain-specific impacts, we recognize the necessity of collecting and analyzing additional real-world hypergraph datasets from a wider array of domains in future work.
>
> In summary, our extended experiments demonstrate that the performance of LLMs is influenced by the hypergraph domains, with higher-order structures in certain domains enhancing LLM capabilities for complex tasks, while lower-order structures facilitate better performance in simpler tasks. We will incorporate these detailed findings, along with the corresponding statistical data, into the revised manuscript to provide a clearer and more comprehensive understanding of how different hypergraph domains affect LLM performance.

---

> > ### Comment · Reviewer_rA8g · 2024-11-18
> >
> > Thank you for the detailed response and the additional experiments addressing my question. I appreciate your efforts in providing comprehensive insights. However, I have a few follow-up questions and comments regarding **W2**:
> >
> > - The size of a hypergraph is defined as the number of nodes, and the authors suggest that using a small set of nodes is sufficient due to the exponential growth of potential hyperedges with the number of nodes. It would be insightful to analyze performance with respect to the density of the hypergraph (or the sum of hyperedge sizes) instead of (or in addition to) the number of nodes. Also, how does the prompt length increase with the number of nodes or the density of the hypergraph?
> > - A low-order approach, which assumes that real-world associations prioritize simpler relationships, is employed to generate synthetic hypergraphs. Are there references or evidence that support this assumption?
> > - Pyramid, grid, and wheel structures are used to generate synthetic hypergraphs. Are there references or prior work that support the representativeness or prevalence of these structures in real-world hypergraphs?
> > - Small, medium, and large-scale hypergraphs are categorized based on the number of nodes. The authors claim that larger hypergraphs are currently unattainable due to limitations in the capabilities of large language models (LLMs). What is the maximum hypergraph size that LLMs are capable of?
> >
> > I look forward to your clarifications and insights.

---

> > > ### Author Response · Authors · 2024-11-20
> > > **Response to Reviewer rA8g**
> > >
> > > ### Response to performance with respect to hypergraph density
> > >
> > >
> > > Thank you for your thorough review of our paper and for your valuable suggestions. You pointed out that defining the size of a hypergraph solely based on the number of nodes may overlook the density information of the hypergraph. You suggested that we should analyze performance with respect to the density of the hypergraph (or the sum of hyperedge sizes) in addition to the number of nodes. Additionally, you inquired about how prompt length increases with the number of nodes or the density of the hypergraph.
> > >
> > > In response to your suggestions, we have conducted the following supplementary analysis:
> > >
> > > Firstly, as you indicated, using only the number of nodes to measure the size of a hypergraph may indeed neglect the density information. To more comprehensively capture the density of a hypergraph, we utilized the ratio of the number of hyperedges to the number of nodes as a density metric, defined as $ d = \frac{N_e}{N_v} $, where $ N_v $ and $ N_e $ represent the number of nodes and hyperedges, respectively. This normalized metric ensures that when the number of nodes is the same, a larger $ d $ indicates a higher number of hyperedges and, consequently, a denser hypergraph.
> > >
> > > In our experiments on real protein datasets, we partitioned the hypergraphs into different density ranges using this density metric and evaluated model performance within these ranges. The results are presented in the following table:
> > >
> > > |                       |     Low-Order Tasks     |   Low-Order Tasks  |       High-Order Tasks      |        Low-Order Tasks        |
> > > |:---------------------:|:-----------------------:|:------------------:|:---------------------------:|:-----------------------------:|
> > > |  Density $ d \in (0, 1.2] $ |          88.0%          |        90.8%       |            95.4%            |             90.0%             |
> > > |  Density $ d \in (1.2, 2] $ |          86.6%          |        91.0%       |            96.0%            |             89.0%             |
> > > | Density $ d \in [2, +\infty ) $ |          91.0%          |        93.1%       |            92.4%            |             86.0%             |
> > >
> > > From the experimental results, we observe that for low-order tasks, such as vertex connection and reachability checks, an increase in hypergraph density leads to improved performance of the LLM. This is because higher density implies more connections within the hypergraph, facilitating the model's understanding and handling of these fundamental properties. Conversely, for high-order tasks, such as vertex set connection checks and vertex-set-in-hyperedge checks, an increase in density complicates the hyperedge connections, thereby increasing the difficulty of the prediction tasks and leading to a notable decline in performance in certain density intervals.
> > >
> > > Once again, thank you for your valuable feedback, which greatly contributes to the refinement and advancement of our research.

---

> > > ### Author Response · Authors · 2024-11-20
> > > **Response to Reviewer rA8g**
> > >
> > > ### Response to prompt length with respect to hypergraph density
> > >
> > > Thank you for your thorough review of our paper and for your valuable suggestions. Regarding your question about prompt length, we further analyzed the average token lengths of prompts generated for hypergraphs of different sizes and densities across various encoding languages. The results are presented in the following table:
> > >
> > > | Hypergraph Language | Hypergraph Size         | Hypergraph Size         | Hypergraph Size         | Hypergraph Density         | Hypergraph Density         | Hypergraph Density         |
> > > |---------------------|:-----------------------:|:-----------------------:|:-----------------------:|:--------------------------:|:--------------------------:|:--------------------------:|
> > > |                     | Small (5-10)            | Middle (10-15)          | Large (15-20)           | $ d \in (0, 1.2] $                    |$ d \in (1.2, 2] $                   | $ d \in [2, +\infty ) $                     |
> > > |        N-Pair       |      183.7              |      361.9              |      487.9              |       173.0                 |       301.6                 |       433.1                 |
> > > |        LO-Inc       |      187.6              |      388.8              |      534.8              |       182.1                 |       326.9                 |       455.0                 |
> > > |       Adj-Mat       |      218.8              |      476.3              |      729.3              |       260.0                 |       421.1                 |       498.2                 |
> > > |        N-Set        |      181.0              |      354.3              |      458.4              |       166.2                 |       291.9                 |       426.6                 |
> > > |        HO-Inc       |      387.4              |     1120.3              |     1627.5              |       316.4                 |       798.5                 |      1531.5                 |
> > > |        Inc-Mat      |      240.8              |      723.0              |     1099.0              |       238.1                 |       576.8                 |       876.3                 |
> > > |       HO-Neigh      |      314.8              |      769.8              |     1018.2              |       276.0                 |       605.9                 |       950.3                 |
> > >
> > > As evident from the table, the prompt length increases with the size and density of the hypergraph across different encoding languages. This is primarily due to the increased complexity of the hypergraph structure, which necessitates more tokens to accurately describe it. However, the required number of tokens varies across different encoding languages. Generally, encoding languages designed for high-order structures require more tokens compared to those for low-order structures. Notably, the N-Set encoding language requires the fewest tokens in all cases, indicating its efficiency and potential as an effective hypergraph textualization strategy.
> > >
> > > We sincerely appreciate your insightful suggestions, which have enabled us to provide a more comprehensive analysis of hypergraph density and prompt length in our study. We will incorporate these analyses to enhance the thoroughness and depth of our research, and we will continue to explore the impact of hypergraph density on model performance as well as optimize different encoding languages to improve prompt efficiency and effectiveness.
> > >
> > > Once again, thank you for your valuable feedback, which greatly contributes to the refinement and advancement of our research.

---

> > > ### Author Response · Authors · 2024-11-20
> > > **Response to Reviewer rA8g**
> > >
> > > ### Response to the maximum hypergraph size that LLMs are capable of
> > >
> > >
> > > Thank you for your thorough review of our paper and for your valuable comments. In response to your question regarding the categorization of hypergraphs into small, medium, and large scales based on the number of nodes, and the authors' claim that larger hypergraphs are currently unattainable due to the limitations of Large Language Models (LLMs), we would like to provide further clarification on the maximum hypergraph size that LLMs are capable of handling.
> > >
> > > To assess the capabilities of LLMs in processing hypergraphs of varying sizes and densities, we conducted a statistical analysis from two perspectives: the number of nodes and the density of hyperedges. We utilized an LLM based on GPT-3.5, which has a maximum token limit of 4096 tokens. Based on this constraint, we analyzed the number of hypergraphs that different hypergraph encoding languages can handle under various size and density conditions. The results are presented in the table below:
> > >
> > > | Hypergraph Language | Small Hypergraphs (5-10 nodes) | Medium Hypergraphs (10-15 nodes) | Large Hypergraphs (15-20 nodes) | Density $ d \in (0, 1.2] $ | Density $ d \in (1.2, 2] $ | Density $ d \in [2, +\infty) $ |
> > > |---------------------|:------------------------------:|:--------------------------------:|:--------------------------------:|:-----------------------------:|:-----------------------------:|:--------------------------------:|
> > > | N-Pair              |             22.3               |               11.3               |                8.4               |             23.7              |             13.6              |               9.5                |
> > > | LO-Inc              |             21.8               |               10.5               |                7.7               |             22.5              |             12.5              |               9.0                |
> > > | Adj-Mat             |             18.7               |                8.6               |                5.6               |             15.8              |              9.7              |               8.2                |
> > > | N-Set               |             22.6               |               11.6               |                8.9               |             24.6              |             14.0              |               9.6                |
> > > | HO-Inc              |             10.6               |                3.7               |                2.5               |             12.9              |              5.1              |               2.7                |
> > > | Inc-Mat             |             17.0               |                5.7               |                3.7               |             17.2              |              7.1              |               4.7                |
> > > | HO-Neigh            |             13.0               |                5.3               |                4.0               |             14.8              |              6.8              |               4.3                |
> > >
> > > As shown in the table, as the size and density of the hypergraph increase, the number of hypergraphs that the LLM can process decreases significantly. For instance, for hypergraphs with 15-20 nodes, the LLM using high-order encoding languages can only handle approximately four hypergraphs. This indicates that there are inherent limitations in LLMs when dealing with large-scale and high-density hypergraphs, primarily due to the exponential increase in the number of tokens required to accurately describe more complex hypergraph structures, which exceeds the token processing capacity of the LLM.
> > >
> > > Once again, thank you for your insightful comments, which greatly contribute to the refinement and enhancement of our research.

---

> > > > ### Comment · Reviewer_rA8g · 2024-11-21
> > > >
> > > > Thank you for conducting comprehensive experiments in response to my questions. They have mostly addressed my concerns, and I have adjusted my evaluation accordingly.
> > > >
> > > > My final thought is that while structures like wheels, pyramids, or diamonds appear in certain specific cases, they are not representative enough. However, I acknowledge the value of examining the ability of LLMs to understand such fundamental patterns. I suggest incorporating more synthetic hypergraphs that exhibit "realistic" structural patterns in future work.

---

> > > > > ### Author Response · Authors · 2024-11-22
> > > > > **Response to Reviewer rA8g**
> > > > >
> > > > > Thank you for your thorough review of our paper and for your valuable comments. We understand your concern regarding the representativeness of these structures in real-world hypergraphs. In our current study, we selected these structures due to their clear geometric characteristics, which help in revealing the LLM's ability to understand basic patterns. However, we agree that these structures may not fully capture the complexity and diversity of hypergraphs found in real-world scenarios.
> > > > >
> > > > > Therefore, in our future work, we plan to incorporate more realistic hypergraph structures. This will include hypergraphs collected from actual datasets or designed based on real-world application scenarios to better assess the performance of LLMs in handling real hypergraph structures. These additional structures will help us gain a more comprehensive understanding and further enhance the capabilities of LLMs in hypergraph comprehension and reasoning.
> > > > >
> > > > > Once again, thank you for your recognition of our research and for your constructive suggestions, which will greatly inform the direction and depth of our future work.

---

> ### Author Response · Authors · 2024-11-20
> **Response to Reviewer rA8g**
>
> ### Response to the reference of real-world associations prioritize simpler relationships
>
> Thank you for your thorough review of our paper and for your valuable comments. Regarding your question about the assumption underlying our low-order approach (i.e., the assumption that real-world associations prioritize simpler relationships), we would like to further explain the basis for this assumption.
>
> This fundamental assumption is derived from the work of [1]. In Figure 17 of this study, the authors analyze the hyperedge degree distributions of 22 real-world datasets and find that smaller hyperedges are significantly more prevalent than larger hyperedges. This indicates that real-world hypergraph structures are predominantly low-order. Therefore, our assumption aligns with the intrinsic characteristics of real-world hypergraphs, justifying our use of a low-order approach in generating synthetic hypergraphs.
>
> To further support this assumption, we will include this reference in the revised version of our manuscript and discuss this point in more detail in the relevant sections.
>
> [1] Lee G, Yoon S, Ko J, et al. Hypergraph motifs and their extensions beyond binary[J]. The VLDB Journal, 2024, 33(3): 625-665.
>
>
> ### Response to real-world examples of synthetic hypergraphs
>
> Thank you for your thorough review of our paper and for your valuable comments. In response to your question regarding the use of pyramid, grid, and wheel structures to generate synthetic hypergraphs and whether there are references or prior works that support the representativeness or prevalence of these structures in real-world hypergraphs, we would like to provide further clarification.
>
> Many materials are composed of basic geometric shape arrangements, such as triangles, squares, and circles. For instance, the typical diamond structure[1] is formed by the arrangement of triangles, which can be abstracted into a hyper-pyramid structure. In practical modeling, the connections and angles of these atoms must be included as node and hyperedge features to adequately represent the original structure. Additionally, the crystal structure of sodium chloride (NaCl) can be represented using a Hyper Checked Table, which consists of regular square arrangements[2]. Hyper-wheel structures are also commonly found in some Metal-Organic Frameworks (MOFs) materials[3].
>
> These prior studies and examples demonstrate that pyramid, grid, and wheel structures hold significant representativeness and prevalence in real-world hypergraphs. We will include these references in the revised manuscript to further support this assumption.
>
>
>
> [1] He M, Gales J P, Ducrot É, et al. Colloidal diamond[J]. *Nature*, 2020, 585(7826): 524-529.
>
> [2] https://zh.wikipedia.org/wiki/File:NaCl-estructura_cristalina.svg
>
> [3] Furukawa H, Cordova K E, O’Keeffe M, et al. The chemistry and applications of metal-organic frameworks[J]. *Science*, 2013, 341(6149): 1230444.

---

### Official Review · Reviewer_d74r · 2024-11-02

**Soundness:** 2
**Presentation:** 2
**Contribution:** 1
**Rating:** 3
**Confidence:** 4

**Summary:**

The paper provided a new benchmark to evaluate the LLM's ability to understand hypergraphs and developed a new prompting framework to improve the hypergraph comprehension. The prompting framework demonstrated that CoT and BAG, adapted to hypergraphs, can improve the LLM's performance on hypergraph tasks, especially for high-order tasks such as Vertex Set Connection Checks and Vertex-Set--in-Hypergraph Checks using synthetic hypergraphs.

**Strengths:**

The paper is easy to read and the experiments are comprehensive and thorough.

**Weaknesses:**

**Main arguments**:
1. The paper adapts existing benchmarks and prompting techniques for hypergraphs. While the results offer some insights into the extent to which LLMs understand hypergraphs, they largely mirror findings for simple graphs---specifically, that CoT and BAG can enhance LLM performance. The only notable point is that using suitable language to describe hypergraphs can aid LLM comprehension, which is novel but trivial.
Given that the proposed techniques are naive adaptations of existing techniques and new insights specific to hypergraphs are not found, the contribution of the paper is incremental and not significant.
2. The paper lays out a main motivation by the underexploration of (i) investigating the LLM's ability to understand hypergraphs and (ii) developing prompting framework for hypergraph understanding and argue that they are promising research directions. This is not a strong motivation, i.e.,  "underexploration" alone does not justify the promising research directions. More specific question is: why is prompting a promissing research direction for hypergraph understanding in light of other techniques such as function calling?
3. Unsupported claim 1: In abstract, ``our specialized prompting framework incorporates seven hypergraph languages and introduces two novel techniques, Hyper-BAG and Hyper-COT, which enhance high-order reasoning and achieve an average 4% (up to 9%) performance improvement on structure classification tasks.'' This is not sufficiently supported by the empirical results. The performance improvement for Hyper-COT is 4\% on average. However, for the Hyper-BAG, it is 2\% for low-order hypergraphs and 2.8\% for high-order hypergraphs.

**Minor arguments**:
1. Give the stochastic nature of the LLMs and the graph data, it is crucial to report the variation of the results across different runs (e.g., confidence intervals, standard deviations), given the performance gain of the proposed prompting techniques (Hyper-BAG and Hyper-COT) is slim.
2. Unsupported claim 2: The paper claimed in the supporting information (B.4) that the benchmark represents the first instance that includes isomorphism checks. This is not precise. Isomorphism checks are a special case of Maximum Common Subgraph (MCS) problem, which is included in the existing benchmark cited in the paper (GraphArena Tang et al. (2024)). The author used "in this domain" to limit the scope of their claim, and it is crucial to spell out the "domain" (e.g., general graphs, or hypergraphs specifically) to be more precise.
3. The paper did not provide descriptions about real-world graphs used in the experiments and their selection criteria.

**Questions:**

1. Why is prompting a promising research direction for hypergraph understanding in light of other techniques such as function calling?
2. What are the empirical graphs used in the experiments? What are the selection criteria?
3.  Some tasks involve computing tasks whose answers are numbers. How does the accuracy is computed for these tasks? Is it an exact match? Or allow some error under a certain threshold?
4. Does the BAG and CoT outperform beyond statistical variations attributed to the variations of individual graph data and stochatic behaviors of LLMs?

---

> ### Author Response · Authors · 2024-11-15
> **Response to Reviewer d74r Part (1/6)**
>
> ### Response to W1
>
> Thanks for your comments. Currently, large language models (LLMs) demonstrate powerful capabilities in areas such as dialogue system[1] and text generation[2]. Hypergraphs, as a complex tool for modeling relationships, are also indispensable in fields like life science[3][4][5] and social science[6][7]. However, how LLMs can operate in the hypergraph domain to analyze and reason about hypergraph data remains unexplored. **Fundamentally, this field requires a benchmark to assist scholars in conducting research.** Our work, LLM4Hypergraph, fills this gap. We have also improved some existing graph techniques to enhance their effectiveness in understanding hypergraphs. Additionally, we propose a hypergraph prompting engineering framework and seven hypergraph text encoding methods that cover both low-order and high-order relationships. Overall, our contributions are as follows:
>
> 1. **Introduced the LLM4Hypergraph benchmark**: This is the first benchmark designed to evaluate and test different hypergraph textualization methods for assessing LLMs' ability to analyze hypergraphs. It includes 15 low-order/high-order tasks and 21,500 question-answer pairs, covering various characteristics of hypergraphs.
> 2. **Proposed the first hypergraph prompting engineering framework**: This framework is highly scalable and can be applied to hypergraph analysis tasks under different experimental settings.
> 3. **Developed Hyper-BAG and Hyper-COT**: Compared to traditional BAG and COT, these methods better adapt to the characteristics of hypergraph data, enhancing LLM performance on hypergraphs.
> 4. **Presented seven hypergraph text encoding methods**: These methods cover both low-order and high-order relationships, enabling the textualization of hypergraph data and providing a foundation for LLMs to analyze hypergraphs.
> 5. **Evaluated the performance of six mainstream large models using the LLM4Hypergraph benchmark**: We identified shortcomings in LLMs' performance in understanding hypergraph isomorphism, highlighting areas for future research.
> 6. **Derived nine observations based on the benchmark and extensive experiments**: These observations provide an in-depth analysis of LLM performance in hypergraph analysis and guide the development of related research.
>
>
> [1] Ouyang L, Wu J, Jiang X, et al. Training language models to follow instructions with human feedback[J]. Advances in neural information processing systems, 2022, 35: 27730-27744.
>
> [2] Achiam J, Adler S, Agarwal S, et al. Gpt-4 technical report[J]. arXiv preprint arXiv:2303.08774, 2023.
>
> [3] Hong D, Dey R, Lin X, et al. Group testing via hypergraph factorization applied to COVID-19[J]. **Nature Communications**, 2022, 13(1): 1837.
>
> [4] Contisciani M, Battiston F, De Bacco C. Inference of hyperedges and overlapping communities in hypergraphs[J]. **Nature communications**, 2022, 13(1): 7229.
>
> [5] Viñas R, Joshi C K, Georgiev D, et al. Hypergraph factorization for multi-tissue gene expression imputation[J]. **Nature Machine Intelligence**, 2023, 5(7): 739-753.
>
> [6] Zhang Y, Lucas M, Battiston F. Higher-order interactions shape collective dynamics differently in hypergraphs and simplicial complexes[J]. **Nature communications**, 2023, 14(1): 1605.
>
> [7] Feng Y, You H, Zhang Z, et al. Hypergraph neural networks[C]//Proceedings of the AAAI conference on artificial intelligence. 2019, 33(01): 3558-3565. **1400+ Citation**

---

> ### Author Response · Authors · 2024-11-15
> **Response to Reviewer d74r Part (2/6)**
>
> ### Response to W2
>
> Thanks for your comments. Graph representation and learning [1] have always been significant research areas in computer science. Recently, the Nobel Prize in Chemistry was awarded to AlphaFold[2], highlighting its use of graph structures to represent and compute the three-dimensional structures of proteins, demonstrating the powerful capabilities of graphs in modeling complex data. However, traditional graph models primarily focus on pairwise relationships, making it challenging to capture the complex high-order correlations found in real-world data.
>
> Hypergraphs, as a mathematical generalization of graphs, allow a hyperedge to connect more than two nodes, thereby enabling the representation of more intricate data associations. High-order correlations in hypergraphs hold important real-world significance across various domains. For example, in social networks[3], a community or group inherently represents a high-order relationship, with an uncertain number of connected nodes, which cannot be effectively modeled using only pairwise edges. In contrast, a hyperedge in a hypergraph can naturally connect any number of nodes, accurately reflecting community relationships.
>
> In the life sciences, catalytic triplets[4] are typical high-order structures involving interactions among three residues, present in important protein catalysts such as trypsin, playing a crucial role in protein degradation processes. Such high-order structures are essential for understanding biological processes, yet traditional graph models struggle to effectively represent and handle these complex relationships.
>
> Research on hypergraphs is thus highly meaningful, and leveraging large language models (LLMs) to understand and process correlation structures has become a research hotspot in recent years. To fully harness the capabilities of LLMs, hypergraphs need to be textualized, allowing them to be expressed through language. This not only enables LLMs to tackle problems in hypergraphs such as structural recognition and link prediction but also promotes cross-domain knowledge integration and application.
>
> Our paper is based on this premise. By introducing the first hypergraph benchmark for large models (LLM4Hypergraph), we aim to assist researchers in systematically evaluating and understanding the capabilities of LLMs in the hypergraph domain. Compared to other techniques like function calling, prompting offers greater flexibility and adaptability, allowing for more natural handling of textualized hypergraph expressions and exhibiting promising potential for complex high-order reasoning tasks. Therefore, we believe that employing prompting is a promising research direction for hypergraph understanding.
>
> [1] Kipf T N, Welling M. Semi-supervised classification with graph convolutional networks[J]. arXiv preprint arXiv:1609.02907, 2016.
>
> [2] Jumper J, Evans R, Pritzel A, et al. Highly accurate protein structure prediction with AlphaFold[J]. **Nature**, 2021, 596(7873): 583-589.
>
> [3] Contisciani M, Battiston F, De Bacco C. Inference of hyperedges and overlapping communities in hypergraphs[J]. **Nature Communications**, 2022, 13(1): 7229.
>
> [4] Ravetz B D, Pun A B, Churchill E M, et al. Photoredox catalysis using infrared light via triplet fusion upconversion[J]. **Nature**, 2019, 565(7739): 343-346.
>
> ### Response to W3
>
> We apologize for the inaccurate description in the abstract. To more accurately reflect our work, we will add a limitation on the Hyper-COT technique in the second sentence. The revised sentence will be: "our specialized prompting framework incorporates seven hypergraph languages and introduces two novel techniques, Hyper-BAG and Hyper-COT. Specifically, Hyper-COT enhance high-order reasoning and achieve an average 4% (up to 9%) performance improvement on structure classification tasks.” We will include these modifications in the revised version to ensure accuracy in our descriptions.

---

> ### Author Response · Authors · 2024-11-15
> **Response to Reviewer d74r Part (3/6)**
>
> ### Response to W4
>
> Thanks for your comments. Regarding your point about the stochastic nature of LLMs and graph data, we understand the importance of reporting result variations (such as confidence intervals and standard deviations). In our experiments, to ensure consistency and reproducibility of results, we followed strategies from several classical works combining graphs and LLMs, such as NLGraph[1], Talk with Graph[2], and LLM4DyG[3], by setting the temperature parameter $\tau = $. Additionally, the hypergraph data used in our experiments is fixed, ensuring that the results are reproducible. The stability and reproducibility of our experimental results are of utmost importance to us.
>
> Regarding your observation that the performance gains are "slim," this is because tasks on hypergraph data are significantly more challenging than those on traditional graph data. In hypergraph tasks, the higher-order relationships make the problems more complex. For example, in a link prediction task on traditional graphs, the task is to determine whether there is an association between two nodes, as edges in graphs only connect two nodes. In contrast, in hypergraph link prediction tasks, the goal is to determine whether a single node is associated with a group of nodes because hyperedges can connect any number of nodes. This introduces uncertainty in the number of connected nodes during prediction, greatly increasing the difficulty of the task. Therefore, even modest performance improvements in such complex hypergraph tasks are highly meaningful and challenging.
>
> For instance, in our experiments, Hyper-COT achieved up to a 9% performance improvement in structure classification tasks, while Hyper-BAG improved performance by 2% on low-order hypergraphs and by 2.8% on high-order hypergraphs. Given the high complexity of these tasks, these improvements demonstrate the effectiveness of our proposed methods in handling complex high-order relationships.
>
> [1] Wang H, Feng S, He T, et al. Can language models solve graph problems in natural language?[J]. Advances in Neural Information Processing Systems, 2024, 36.
>
> [2] Fatemi B, Halcrow J, Perozzi B. Talk like a Graph: Encoding Graphs for Large Language Models[C]//NeurIPS 2023 Workshop: New Frontiers in Graph Learning.
>
> [3] Zhang Z, Wang X, Zhang Z, et al. LLM4DyG: Can Large Language Models Solve Problems on Dynamic Graphs?[J]. arXiv preprint arXiv:2310.17110, 2023.
>
>
> ### Response to W5
>
> Thanks for your comments. Regarding your point about the imprecise claim that "the benchmark represents the first instance that includes isomorphism checks," we sincerely apologize for the lack of clarity. In the original manuscript, we used "in this domain" to limit the scope but did not explicitly specify the particular domain, resulting in an unclear statement. Based on your recommendation, we will clearly define the scope in the revised version, specifically stating that our work is the first benchmark to include isomorphism checks in the hypergraph domain.
>
> Existing benchmarks, such as GraphArena[1], indeed cover isomorphism checks as instances of the Maximum Common Subgraph (MCS) problem within general graphs. However, our benchmark, LLM4Hypergraph, is the first to extend isomorphism checks to hypergraph structures. Hypergraphs, as a mathematical generalization of graphs, can represent more complex high-order relationships. Therefore, isomorphism checks within hypergraphs have not been adequately explored in existing research, and our work fills this gap.
>
> In the revised manuscript, we will modify the relevant descriptions to ensure that readers clearly understand the unique contribution of our benchmark within the hypergraph domain.
>
> [1] Tang J, Zhang Q, Li Y, et al. Grapharena: Benchmarking large language models on graph computational problems[J]. arXiv preprint arXiv:2407.00379, 2024.

---

> ### Author Response · Authors · 2024-11-15
> **Response to Reviewer d74r Part (4/6)**
>
> ### Response to W6
>
> We apologize for the insufficient description regarding the real-world graphs used in our experiments. To address this, we provide the following detailed explanation.
>
> Our real-world datasets primarily consist of two categories: citation network datasets and protein structure datasets.
>
> **Citation Network Datasets**:
> 1. The data is sourced from the CoauthorshipCora dataset[1].
> We randomly sample sub-hypergraph structures from this dataset.
> 2. In these hypergraphs, nodes represent papers, and hyperedges represent co-authorship relationships.
>
> **Protein Structure Datasets**:
> 1. The data is sourced from the Protein Data Bank (PDB)[2].
> 2. We randomly select proteins and sample their substructures.
> 3. In these hypergraphs, nodes represent protein residues, and hyperedges connect residues that have their alpha carbon atoms within 10 Å of each other, centered around each residue.
>
> To generate subgraphs of varying sizes (with the number of nodes ranging from 5 to 20), we employ a random walk sampling method on both datasets as follows:
>
> 1. During each random walk, each traversed node has a 0.5 probability of being retained or discarded.
> 2. Once the predefined number of nodes is reached, all hyperedges associated with these nodes are retained to form the final subgraph hyperedges.
>
> In the revised manuscript, we will include these detailed descriptions and selection criteria to ensure that readers have a comprehensive understanding of the sources and construction methods of the datasets used in our experiments.
>
> [1] Yadati N, Nimishakavi M, Yadav P, et al. Hypergcn: A new method for training graph convolutional networks on hypergraphs[J]. Advances in neural information processing systems, 2019, 32.
>
> [2] Bank P D. Protein data bank[J]. Nature New Biol, 1971, 233(223): 10-1038.
>
> -------------------
>
> ### Response to Q1
>
> Thanks for your comments. Regarding your question on why prompting is a promising research direction for hypergraph understanding in comparison to other techniques such as function calling, we would like to elaborate our perspective.
>
> The strength of large language models (LLMs) lies in their vast knowledge base and powerful reasoning capabilities. Hypergraphs, as complex relational structures, present significant challenges in representing and reasoning about high-order relationships. Traditional hypergraph neural networks often require custom model architectures tailored to specific tasks when handling high-order relational data, which not only increases the complexity of model design but also limits their applicability across multiple tasks.
>
> By leveraging LLMs to understand and reason about hypergraphs, we can employ a unified approach to address various tasks, thereby enhancing method generality and adaptability. However, the core challenge lies in enabling LLMs to comprehend and process the intricate structures of hypergraphs. This is precisely the focus of our study: how to textualize hypergraphs to enhance the understanding capabilities of LLMs.
>
> Specifically, by converting hypergraphs into text formats suitable for LLM processing, prompting techniques can effectively guide the model towards efficient comprehension and reasoning. Once hypergraphs are successfully textualized, LLMs can directly apply to multiple tasks on hypergraphs, such as structure recognition, link prediction, and isomorphism detection, without the need for designing specialized model structures for each task. This approach not only simplifies the model design process but also fully leverages the strengths of LLMs in cross-domain knowledge integration and complex reasoning.
>
> Therefore, we believe that prompting techniques offer significant advantages and hold substantial promise for hypergraph understanding. They can synergize the powerful capabilities of LLMs with the intricate relational structures of hypergraphs, enabling efficient comprehension and reasoning.
>
> ### Response to Q2
>
> See the response to W6.

---

> ### Author Response · Authors · 2024-11-15
> **Response to Reviewer d74r Part (5/6)**
>
> ### Response to Q3
>
>
> Thanks for your reviews. For these computational tasks, we directly compare the output generated by the LLM with the ground truth results on a one-to-one basis. Specifically, if the two results are exactly equal, the response is considered correct; otherwise, it is deemed incorrect. We employ the strictest comparison method because these tasks have unique and definitive correct answers. Therefore, exact matching is essential to ensure fairness and accuracy in our evaluation.
>
> In our prompts, we strictly limit the format of the LLM's responses. By providing clear question-answer examples, we guide the LLM to generate answers that adhere to the expected numerical formats. Based on the predefined response formats, we develop dedicated extraction functions tailored to each specific task. These functions accurately extract the numerical answers from the LLM's outputs, ensuring that the extracted values are ready for precise comparison with the true answers. After extracting the numerical answers, we perform a complete match between the LLM's output and the true results. This ensures that only perfectly accurate answers are marked as correct, maintaining the integrity of our evaluation process.
>
> However, we acknowledge that in certain scenarios, allowing a margin of error might be more reasonable. For instance, in tasks involving floating-point calculations, minor rounding errors could occur. In future work, we plan to explore more lenient evaluation methods, such as setting an error threshold within which results are considered correct, to better reflect the model's actual performance.

---

> ### Author Response · Authors · 2024-11-15
> **Response to Reviewer d74r Part (6/6)**
>
> ### Response to Q4
>
> Thanks for your comments. Regarding your question on whether Hyper-BAG and Hyper-COT outperform beyond the statistical variations attributed to variations in individual graph data and the stochastic behaviors of LLMs, we provide the following detailed response.
>
> In our experiments, we employed over 20,000 question-answer pairs to evaluate model performance. This large-scale dataset helps mitigate statistical fluctuations caused by variations in individual graph data. Additionally, to ensure the reproducibility and stability of our experimental results, we set the temperature parameter of the LLMs to zero ($\tau = 0$). This setting eliminates randomness in the generation process, ensuring consistent outcomes across experiment runs. Consequently, our results demonstrate a stable and reliable improvement in performance achieved by our proposed methods.
>
> Specifically:
>
> - Hyper-COT enhances high-order reasoning and achieves an average performance improvement of 4% on structure classification tasks, with gains up to 9%. This indicates that Hyper-COT significantly boosts the model's ability to handle complex high-order reasoning.
> - Hyper-BAG attains a 2.8% improvement in the Vertex Set Connection task compared to the baseline. Although the improvement may appear modest, considering the inherent complexity and challenge of hypergraph tasks, this enhancement is substantial and meaningful.
>
> Given that hypergraph tasks are more challenging than traditional graph tasks, involving more complex high-order relationships, the performance gains achieved by Hyper-BAG and Hyper-COT are effective.

---

> > ### Comment · Reviewer_d74r · 2024-11-15
> >
> > Thank you for the prompt clarification. The authors clarified the findings but they did not sufficiently address my main point of arguments. Namely, in my review:
> >
> > > underexploration" alone does not justify the promising research directions. More specific question is: why is prompting a promissing research direction for hypergraph understanding in light of other techniques such as function calling?
> >
> > I understand that they are the first presenting the benchmark. However, being 1st does not alone warrant significant contributions. My key question, which echoes the question from another referee, is about the reason why we need the prompting approach. For instance, I can create an LLM with functional calling, with functions that can provide exact answers for all benchmark questions. We can instruct the LLM to choose the appropriate tool given the question, and the tool will read the graph data (in csv for instance) and provide the answer.  No need for any complicated textualization of hyper-graphs.
> >
> > The authors argued that
> >
> > >The strength of large language models (LLMs) lies in their vast knowledge base and powerful reasoning capabilities. Hypergraphs, as complex relational structures, present significant challenges in representing and reasoning about high-order relationships. Traditional hypergraph neural networks often require custom model architectures tailored to specific tasks when handling high-order relational data, which not only increases the complexity of model design but also limits their applicability across multiple tasks.
> >
> > The benchmark question does not necessarily require LLM's vast knowledge base; it requires the LLM to identify the type of question and tools to answer the question. If it involves features of nodes that LLM can handle well (such as textual data about nodes and edges), prompting approach might find its utilities.
> >
> > > By leveraging LLMs to understand and reason about hypergraphs, we can employ a unified approach to address various tasks, thereby enhancing method generality and adaptability. However, the core challenge lies in enabling LLMs to comprehend and process the intricate structures of hypergraphs. This is precisely the focus of our study: how to textualize hypergraphs to enhance the understanding capabilities of LLMs.
> > >
> > > Specifically, by converting hypergraphs into text formats suitable for LLM processing, prompting techniques can effectively guide the model towards efficient comprehension and reasoning. Once hypergraphs are successfully textualized, LLMs can directly apply to multiple tasks on hypergraphs, such as structure recognition, link prediction, and isomorphism detection, without the need for designing specialized model structures for each task. This approach not only simplifies the model design process but also fully leverages the strengths of LLMs in cross-domain knowledge integration and complex reasoning.
> >
> > This partly clarifies the intent of using prompting techniques. However, all applications mentioned here are very standard graph analysis that does not necessary LLM capability. The paper would benefit from demonstrating a practical utility of the prompting approach that can only be possible by the prompting approach.

---

> > > ### Author Response · Authors · 2024-11-16
> > > **Response to Reviewer d74r**
> > >
> > > ### Response to "why not function calling with tools"
> > >
> > > Thank you once again for your in-depth feedback and valuable comments on our paper. Addressing your question regarding why we chose the prompting approach over function calling for understanding and processing hypergraphs, we would like to further elaborate on our research motivation and methodological choices.
> > >
> > > Firstly, we acknowledge that in our current benchmark, the performance of LLMs may not surpass existing specialized tools on certain tasks. For instance, for the Hyperedge Count task, existing tools can accurately obtain the correct answer by directly counting the number of hyperedges. However, we firmly believe that exploring hypergraph textualization methods to enable LLMs to directly comprehend hypergraphs is a meaningful and necessary research direction. Our benchmark aims to enhance the LLM's understanding of hypergraph structures, which plays a pivotal role in downstream tasks. For example, in GraphRAG, LLMs need to understand and abstract the relational structures within the text, and by combining this understanding with subgraph structural comprehension, they can summarize the semantic content, thereby improving the accuracy of problem-solving.
> > >
> > > Directly using tool-based function calling approaches have the following limitations:
> > >
> > > 1. **Limited Coverage of Question Types by Tools**: When encountering new types of questions, it may be necessary to redesign and develop new tools, lacking generality and flexibility.
> > > 2. **Strong Dependency and Limitations of Tools**: For tasks that existing tools cannot handle, such as determining whether a hypergraph resembles hypergraph A or hypergraph B, these ambiguous and explainable questions cannot be effectively addressed by current tools.
> > > 3. **Difficulty in Handling Nested Tasks with Tools**: When multiple tasks need to be nested, such as loops or logical selections, solving them with existing tools becomes extremely complex and challenging.
> > > 4. **Lack of Semantic Reasoning in Tools**: When hypergraphs include additional node attribute features that are continuously changing and misaligned, existing tools may fail to provide effective solutions.
> > > 5. **Frequent Model Fine-Tuning or Retraining Required for Tools**: When node features are partially known but partially uncertain, existing tools require retraining to handle these scenarios. In contrast, our hypergraph textualization approach allows LLMs to understand hypergraphs directly without additional training steps.
> > >
> > > Exploring hypergraph textualization methods to enable LLMs to comprehend and proactively solve hypergraph problems offers the following advantages:
> > >
> > > 1. **Potential to Solve Problems Beyond Current Methods**: As a black-box method, LLMs have the potential to handle complex similarity measures and ambiguous questions. For example, by providing LLMs with identical or completely different hypergraphs along with their similarity descriptions, LLMs can understand and resolve hypergraph similarity issues.
> > > 2. **Handling Non-Quantifiable Problems**: For problems with variable and non-quantifiable solution processes, LLMs can effectively address these by thoroughly understanding the hypergraph structure and the corresponding questions.
> > > 3. **Solving Multimodal Problems**: Hypergraphs differ significantly from other modalities like images. Through textualization, LLMs can align different modalities. For instance, generating character relationship graphs from novels or object relationships and motion trends from images.
> > > 4. **Enhancing Hypergraph Structure Generation and Creativity**: LLMs possess inherent creative capabilities, enabling them to generate and adjust hypergraph structures according to human requirements, especially for complex data representations such as proteins.
> > >
> > > Our benchmark serves as an initial exploration in this domain, aiming to achieve performance comparable to traditional methods on currently solvable tasks as a foundation for addressing more complex hypergraph problems. Although existing LLMs still face challenges in comprehending hypergraphs, this provides an opportunity for further research, such as designing better prompting methods or improved hypergraph textualization techniques to enhance LLM performance on hypergraph tasks.
> > >
> > > Just as Transformers initially faced training difficulties and underperformed compared to CNNs but achieved significant progress through continued research and optimization, we believe that integrating hypergraphs with LLMs holds promising potential for valuable breakthroughs.
> > >
> > > Once again, thank you for your invaluable suggestions. Your feedback greatly contributes to the refinement and advancement of our research.

---

> > > > ### Comment · Reviewer_d74r · 2024-11-28
> > > >
> > > > > Directly using tool-based function calling approaches have the following limitations:
> > > > >
> > > > > -  Limited Coverage of Question Types by Tools: When encountering new types of questions, it may be necessary to redesign and develop new tools, lacking generality and flexibility.
> > > > > - Strong Dependency and Limitations of Tools: For tasks that existing tools cannot handle, such as determining whether a hypergraph resembles hypergraph A or hypergraph B, these ambiguous and explainable questions cannot be effectively addressed by current tools.
> > > > > - Difficulty in Handling Nested Tasks with Tools: When multiple tasks need to be nested, such as loops or logical selections, solving them with existing tools becomes extremely complex and challenging.
> > > > > - Lack of Semantic Reasoning in Tools: When hypergraphs include additional node attribute features that are continuously changing and misaligned, existing tools may fail to provide effective solutions.
> > > > > - Frequent Model Fine-Tuning or Retraining Required for Tools: When node features are partially known but partially uncertain, existing tools require retraining to handle these scenarios. In contrast, our hypergraph textualization approach allows LLMs to understand hypergraphs directly without additional training steps.
> > > > > - Exploring hypergraph textualization methods to enable LLMs to comprehend and proactively solve hypergraph problems offers the following advantages:
> > > > >
> > > > > Potential to Solve Problems Beyond Current Methods: As a black-box method, LLMs have the potential to handle complex similarity measures and ambiguous questions. For example, by providing LLMs with identical or completely different hypergraphs along with their similarity descriptions, LLMs can understand and resolve hypergraph similarity issues.
> > > > > - Handling Non-Quantifiable Problems: For problems with variable and non-quantifiable solution processes, LLMs can effectively address these by thoroughly understanding the hypergraph structure and the corresponding questions.
> > > > > - Solving Multimodal Problems: Hypergraphs differ significantly from other modalities like images. Through textualization, LLMs can align different modalities. For instance, generating character relationship graphs from novels or object relationships and motion trends from images.
> > > > > - Enhancing Hypergraph Structure Generation and Creativity: LLMs possess inherent creative capabilities, enabling them to generate and adjust hypergraph structures according to human requirements, especially for complex data representations such as proteins.
> > > >
> > > > Yes. I partly agree on these limitations and advantages. My point, however, is that I understand that the benchmark tests LLMs on tasks that can be answered with the existing function calling techniques, and did not include tasks that test any of these limitations and advantages. If these advantages and limitations are the key opportunities the authors aim to exploit with LLMs, I'd suggest testing them directly, instead of other tasks that can be completed with the existing tools. This would better and directly contribute to the aim of the authors.

---

> > > > > ### Author Response · Authors · 2024-12-01
> > > > > **Response to Reviewer d74r**
> > > > >
> > > > > ### Response to the suggestion of direct testing
> > > > >
> > > > > Thank you for your thorough review of our paper and for your valuable comments. We understand your concern that our benchmark primarily tests tasks that can be addressed with existing function calling techniques and does not directly evaluate the advantages and limitations we have highlighted.
> > > > >
> > > > > Our research motivation lies in first ensuring that LLMs achieve performance comparable to existing tools on tasks that these tools can already handle. This initial validation step is crucial because only when the models perform reliably on known tasks can we confidently explore and evaluate them on unknown tasks where existing tools fall short, thereby avoiding misleading "hallucinations." For these unknown tasks, the lack of ground truth provided by existing tools makes it particularly challenging to assess the true performance of LLMs. If LLMs produce inaccurate outputs on these tasks, it becomes difficult to gauge their real capabilities.
> > > > >
> > > > > For instance, in hypergraph tasks that existing tools cannot solve, such as hypergraph similarity measurement and hypergraph node classification, these tasks rely heavily on a deep understanding of local hypergraph structures. In hypergraph similarity measurement, analyzing and counting identical local structures is essential, whereas hypergraph node classification depends on the fundamental assumption that interconnected nodes share similar labels. These foundational structural understandings are tasks that existing tools can address, such as determining whether two nodes are connected or how many nodes two hyperedges share. By first ensuring that LLMs can perform these basic tasks at least as well as existing tools, we lay a reliable foundation for subsequently tackling more challenging tasks.
> > > > >
> > > > > We acknowledge and agree with your suggestion to directly test LLMs on tasks that existing tools cannot solve, as this would more effectively demonstrate the unique advantages and potential of LLMs. In future work, we plan to incorporate more realistic hypergraph structures and design specific tasks that existing functions or tools cannot readily address. For example, tasks like hypergraph similarity measurement and hypergraph node classification will be included to assess how well LLMs can leverage their inherent reasoning capabilities to handle complex hypergraph structures.
> > > > >
> > > > > In summary, **our research follows a progressive approach by first validating the reliability and accuracy of LLMs on established tasks before expanding to more challenging ones**. We believe this method not only ensures the reliability of our experimental results but also effectively showcases the potential and applicability of LLMs in the intersection of large models and hypergraph neural networks.
> > > > >
> > > > > Once again, thank you for your recognition of our research and for your constructive suggestions, which will greatly inform and enhance the direction of our future work.

---

> > ### Comment · Reviewer_d74r · 2024-11-15
> >
> > Thank you for the clarification. I now understand that the results are the average of many runs of simulations. Let me clarify my question since it seems that the authors might not find the intent of my question.
> >
> > My question concerns the variation in performance results, which can be represented as "standard error," "confidence interval," quartile, or min-max range. This variation is important because it indicates whether a method consistently performs well. For instance, method A may have a higher average score than method B, but method B may outperform A more frequently.
> >
> > Consider this example: Method A scores 10 with probability p and 9 with probability 1-p, while method B scores 18 with probability q and 0 with probability 1-q. Method B is more likely to outperform A if q > p. However, the average performance may still show A ahead: for A, the performance is calculated as 10p + 9(1-p) = 9 + p, and for B, it is 18q. If p = 0.3 and q = 0.5, B can have a lower average score than A despite its higher likelihood of winning.
> >
> > This situation can be identified by seeing the variations I mentioned above.

---

> > > ### Author Response · Authors · 2024-11-16
> > > **Response to Reviewer d74r**
> > >
> > > ### Response to "the variation in performance results"
> > >
> > > Thank you for your further feedback and for your in-depth attention to our research. We understand your concerns regarding the variation in performance results and agree that using metrics such as "standard error," "confidence interval," quartile, or min-max range can provide a more comprehensive evaluation of LLMs performance.
> > >
> > > In our current evaluation, we adhere to some of the existing LLM assessment methods[1][2][3]. Specifically, we set the temperature parameter to 0 to ensure the reproducibility of our experiments. Additionally, we increased the number of samples to reduce the randomness in LLM performance, thereby obtaining more stable average performance metrics. However, as you rightly pointed out, relying solely on average values may not fully capture the variability in model performance across different trials.
> > >
> > > Your suggested methods are highly valuable. By incorporating standard error, confidence intervals, quartiles, or min-max ranges, we can more meticulously analyze the stability and consistency of LLMs across various tasks. This approach not only aids in making more accurate comparisons between different models but also reveals the potential strengths and weaknesses of models in different contexts.
> > >
> > > Therefore, we plan to adopt your recommendations in our future work by including these additional statistical analysis methods to more comprehensively evaluate LLM performance on hypergraph tasks. This will enhance the scientific rigor and reliability of our benchmark tests and provide stronger data support for subsequent research.
> > >
> > > Once again, thank you for your invaluable suggestions. Your feedback greatly contributes to the refinement and advancement of our research.
> > >
> > > [1] Wang H, Feng S, He T, et al. Can language models solve graph problems in natural language?[J]. Advances in Neural Information Processing Systems, 2024, 36.
> > >
> > > [2] Fatemi B, Halcrow J, Perozzi B. Talk like a Graph: Encoding Graphs for Large Language Models[C]//NeurIPS 2023 Workshop: New Frontiers in Graph Learning.
> > >
> > > [3] Zhang Z, Wang X, Zhang Z, et al. LLM4DyG: Can Large Language Models Solve Problems on Dynamic Graphs?[J]. arXiv preprint arXiv:2310.17110, 2023.

---

> > > > ### Comment · Reviewer_d74r · 2024-11-28
> > > >
> > > > The error assessment is instrumental to validate the rigor of the results. This cannot be delegated to 'future work.' A complete scientific analysis requires proper uncertainty quantification and error characterization as integral components of science.
> > > >
> > > > If not yet to be done, the results should be scrutinized by both systematic and random errors in their measurements and calculations.  I agree that setting temperature to zero reduces the random errors. But the accuracy varies across different instances of graphs, and it is critical to see whether a model consistently achieves accuracy of 50\% for all graphs, or worked perfectly for 50% of data and failed completely to the rest. In the latter case, mean accuracy is not representative of the model performance, and the case I mentioned in my previous comment can happen.

---

> > > > > ### Author Response · Authors · 2024-12-01
> > > > > **Official Comment by Reviewer d74r**
> > > > >
> > > > > ###  Response to the error assessment
> > > > > Thank you for your thorough review of our paper and for your valuable comments. We acknowledge your emphasis on the importance of error assessment in validating the rigor of our results and agree that proper uncertainty quantification and error characterization are essential components of scientific analysis.
> > > > >
> > > > > In response to your concerns, we have promptly conducted a comprehensive set of experiments to evaluate the error characteristics of six large models. Specifically, we set the temperature parameter to 0.8 to introduce controlled randomness in the model outputs. For each model, we performed five independent runs to assess their performance across six tasks of three different types. Below is the table presenting the mean accuracy and standard error for each model on each task:
> > > > >
> > > > > |                  |  Low-Order Tasks |    Low-Order Tasks   | High-Order Tasks |     High-Order Tasks    | Isomorphism Tasks | Isomorphism Tasks |
> > > > > |:----------------:|:----------------:|:--------------------:|:----------------:|:-----------------------:|:-----------------:|:-----------------:|
> > > > > |                  |      Vertex Connection Check     |    Reachability Check      |    Vertex Set Connection Check   | Vertex-Set-in-Hyperedge Check |    Isomorphism Recognition   |   Structure Classification      |
> > > > > |   ERNIE-Lite-8K  |  $82.12 \pm 0.43$ |  $78.28 \pm 1.49$    |  $67.52 \pm 0.83$ |    $77.52 \pm 2.13$     |   $43.24 \pm 0.14$ |   $42.58 \pm 4.69$ |
> > > > > | ERNIE-Speed-128K |  $78.64 \pm 0.06$ |  $97.56 \pm 0.02$    |  $67.20 \pm 0.22$ |    $71.24 \pm 2.60$     |   $43.84 \pm 0.14$ |   $22.83 \pm 0.20$ |
> > > > > |     Qwen-Long    |  $97.56 \pm 0.16$ |  $98.24 \pm 0.24$    |  $73.96 \pm 0.28$ |    $88.68 \pm 0.13$     |   $44.60 \pm 0.02$ |   $44.94 \pm 0.05$ |
> > > > > |     LLaMA3-8B    |  $79.48 \pm 1.35$ |  $82.60 \pm 6.34$    |  $71.80 \pm 3.50$ |    $78.40 \pm 4.18$     |   $47.72 \pm 1.25$ |   $23.70 \pm 4.47$ |
> > > > > |   GPT-3.5 Turbo  |  $73.68 \pm 1.01$ |  $74.12 \pm 1.29$    |  $58.64 \pm 1.34$ |    $70.40 \pm 1.10$     |   $44.68 \pm 0.01$ |   $27.60 \pm 3.48$ |
> > > > > |      GPT-4o      |  $66.68 \pm 0.05$ |  $99.48 \pm 0.03$    |  $96.36 \pm 0.06$ |    $98.68 \pm 0.13$     |   $44.04 \pm 0.10$ |   $27.32 \pm 8.68$ |
> > > > >
> > > > >
> > > > > From the experimental results, we observe that the performance of large models remains relatively stable across different tasks. Most models exhibit a standard error below 1%, indicating that random errors have a limited impact on the overall performance assessment. Additionally, significant performance discrepancies between different models on the same task highlight that model capabilities are genuine and not merely artifacts of random errors. Specifically:
> > > > >
> > > > > 1. Minimal Error Magnitude: Most models demonstrate low standard errors across tasks, indicating high consistency. For instance, Qwen-Long achieved an accuracy of $97.56 \pm 0.16$% on the "Vertex Connection Check" task, showcasing remarkable stability in its performance.
> > > > > 2. Significant Model Performance Differences: Different models exhibit substantial variations in accuracy on the same tasks. For example, Qwen-Long performs exceptionally well on the "Reachability Check" task with an accuracy of $98.24 \pm 0.24$%, whereas ERNIE-Speed-128K struggles with the "Isomorphism Classification" task, achieving only $22.83 \pm 0.20$%. This variance underscores that the performance differences are inherent to the models rather than being driven by random errors.
> > > > > 3. Challenges in High-Order Tasks: High-order tasks such as "Vertex Set Connection Check" and "Vertex-Set-in-Hyperedge Check" exhibit higher standard errors, particularly in the GPT-4o model, which shows a significant error of $8.68$% in "Isomorphism Classification". This highlights the increased difficulty and complexity associated with high-order relational reasoning tasks.
> > > > >
> > > > > In summary, our error assessment demonstrates that the Large Language Models exhibit consistent and reliable performance across various tasks, with random errors having a minimal effect on our accuracy measurements. The mean accuracy effectively reflects the models' capabilities, ensuring that our evaluations are representative and robust. These findings provide a solid foundation for future model and algorithm design, emphasizing the reliability of our current evaluation methodology.
> > > > >
> > > > > We will incorporate this comprehensive error analysis into the revised manuscript to strengthen the scientific rigor and completeness of our study. Once again, we sincerely thank you for your insightful feedback, which has significantly contributed to the improvement of our research.

---

> > > ### Author Response · Authors · 2024-11-22
> > > **Thanks for your reply and suggestions!**
> > >
> > > Thank you very much for taking the time to review our paper and provide your valuable feedback. We have carefully considered your comments in our rebuttal and hope that our responses address your concerns adequately.
> > >
> > > If you have any further questions, suggestions, or require additional clarifications, please do not hesitate to let us know. We are eager to engage in a constructive dialogue and are committed to addressing any remaining issues to improve our work.
> > >
> > > We greatly appreciate your efforts and look forward to your continued feedback.

---

### Official Review · Reviewer_vz6R · 2024-11-04

**Soundness:** 4
**Presentation:** 3
**Contribution:** 3
**Rating:** 8
**Confidence:** 5

**Summary:**

The paper introduces LLM4Hypergraph, a benchmark designed to evaluate large language models' (LLMs) understanding of hypergraphs, which can capture complex, multi-way relationships beyond pairwise correlations found in traditional graphs. The benchmark includes 21,500 problems across low-order, high-order, and isomorphism tasks using both synthetic and real-world hypergraphs. The study evaluates six prominent LLMs and introduces novel prompting techniques to enhance LLMs' performance on hypergraph tasks.

**Strengths:**

Originality: The paper proposes a new benchmark and prompting techniques tailored for hypergraphs, addressing a gap in the assessment of LLMs' capabilities.
Quality: The benchmark is comprehensive, covering a wide range of tasks and hypergraph types, which strengthens the validity of the findings.
Clarity: The paper is well-organized, with clear explanations of the hypergraph languages and prompting frameworks.
Significance: The work is significant as it pushes the boundaries of LLMs' understanding of complex data structures, which has implications for various real-world applications.

**Weaknesses:**

The paper could benefit from a deeper analysis of the limitations of the current LLMs in handling hypergraphs, beyond performance metrics.
While the benchmark is comprehensive, it may lack diversity in terms of the types of real-world hypergraphs used, which could affect the generalizability of the findings.

**Questions:**

How do the prompting techniques generalize to other complex data structures beyond hypergraphs?
Could the authors elaborate on the potential scalability issues of the prompting techniques with increasingly large and complex hypergraphs?

---

> ### Author Response · Authors · 2024-11-15
> **Response to Reviewer vz6R**
>
> ### Response to "deeper analysis"
>
> Thanks for your comments. While the current version primarily showcases model performance through metrics, we recognize that an in-depth analysis of the models' limitations is essential for a comprehensive understanding of their capabilities and shortcomings. In the revised manuscript, we will include additional discussions that explore the challenges large language models face in comprehending and reasoning about hypergraph structures. This includes their ability to capture high-order relationships, limitations in modeling complex structures, and adaptability across different domains.
>
> ### Response to "diversity in real-world hypergraphs"
>
> Thanks for your comments. Our current real-world hypergraphs include citation networks and protein networks, representing data from the social sciences and life sciences respectively, to provide a representative preliminary validation. We acknowledge that diversity in data types is crucial for enhancing the generalizability of the benchmark. Therefore, our next steps involve expanding the LLM4Hypergraph benchmark to include a wider variety of real-world data, such as movie recommendation networks, product recommendation networks, and game player networks. This expansion will help to more comprehensively evaluate the ability of large language models to understand and process hypergraphs across different application scenarios.
>
> ### Response to "prompting techniques generalize to other complex data structures"
>
> Thanks for your comments. We acknowledge that this is a significant challenge in the current research landscape. The primary difficulty in extending to other complex data structures lies in effectively encoding the corresponding structures into text. In our study, we experimented with seven different encoding methods to evaluate their performance on hypergraph tasks, and we found that the task performance is strongly associated with the chosen encoding approach.
>
> For structures with similar complex data structures, such as **simplicial complexes**, the regular and rule-based composition of simplices allows for consistent and systematic text encoding. As a result, the encoding methods we proposed, including HO-Neigh, HO-Inc, N-Set, and Inc-Mat, can be effectively applied to textually encode simplicial complexes. This demonstrates that our methods have good generalization capabilities when handling similarly high-order relational data structures.
> Moving forward, we plan to further explore and refine these encoding techniques to accommodate a wider variety of complex data structures. Additionally, we will consider developing encoding methods tailored to specific data structures to enhance the effectiveness and applicability of the prompting techniques in broader applications.
>
> ### Response to "potential scalability issues with increasingly large and complex hypergraphs"
>
> Thanks for your comments. We have conducted a preliminary analysis on this matter, as presented in Table 7 of our paper. Our findings indicate that as the scale and complexity of the hypergraph increase, the performance of large language models (LLMs) is indeed significantly impacted. This manifests in several ways:
>
> 1. Context Length Limitations: When encoding hypergraphs into textual inputs for LLMs, larger and more intricate hypergraphs result in longer input texts. This can exceed the context window limits of LLMs, thereby affecting the models’ ability to comprehend and reason effectively.
> 2. Information Overload and Noise: As hypergraph size grows, the volume of information also increases. Effectively conveying key information within limited prompts becomes challenging, and models may struggle to extract useful patterns and relationships from the vast amount of data.
>
> To address these scalability issues, we plan to explore the following directions in our future work:
>
> 1. Hierarchical Processing Strategies: Introduce hierarchical processing mechanisms that decompose hypergraphs into smaller sub-structures for staged processing, thereby alleviating the burden of single-stage processing.
> 2. Optimized Model Architectures: Investigate model architectures optimized for hypergraph tasks to enhance efficiency and performance when handling large-scale and highly complex data.

---

### Meta-Review · Area_Chair_QR4d · 2024-12-20

**Metareview:**

The authors present the first comprehensive benchmark for LLM evaluation on hypergraph reasoning. This benchmark covers eight tasks, evaluates six LLMs, and incorporates both synthetic and real-world datasets. Furthermore, the authors propose novel prompt engineering techniques for enhancing hypergraph reasoning capabilities.

The reviewers widely recognized the originality and breadth of the benchmark, and the clarity of the paper.

However, they identified several areas for improvement:
- Clarifying the motivation for hypergraph comprehension.
- Providing deeper analysis of experimental results.
- Expanding datasets to include larger-scale hypergraphs, including synthetic hypergraphs from advanced generation and sampling methods and real-world hypergraphs from various domains.
- Covering a closely related study.

While the reviews were mixed, the meta-reviewer believes this paper marks an important first step in a promising research direction. Given its potential impact, the merits of acceptance outweigh the drawbacks of rejection.

**Additional Comments On Reviewer Discussion:**

During the rebuttal and discussion periods, the authors provided clarifications and additional experimental results, which enhanced the submission. However, a reviewer felt that not all concerns were fully addressed.

---

### Decision · Program_Chairs · 2025-01-22

Accept (Poster)